s

# Molecular scale description of interfacial mass transfer in phase separated aqueous secondary organic aerosol

Mária Lbadaoui-Darvas[1], Satoshi Takahama[1], and Athanasios Nenes[1,2]

[1]School of Architecture, Civil and Environmetal Engineering, Swiss Federal Institute of Technology, Lausanne, 1015, Switzerland
[2]Institute of Chemical Engineering Sciences, Foundation for Research and Technology Hellas, Patras, Greece GR-26504

**Correspondence:** Athanasios Nenes (athanasios.nenes@epfl.ch), Satoshi Takahama (satoshi.takahama@epfl.ch)

**Abstract.** Liquid-liquid phase separated (LLPS) aerosol particles are known to exhibit increased CCN activity compared to well mixed ones due to a complex effect of low surface tension and non-ideal mixing. The relation between the two contributions as well as the molecular scale mechanism of water uptake in the presence of an internal interface within the particle is to date not fully understood. Here we attempt to gain understanding in these aspects through steered molecular dynamics simulation studies of water uptake by a vapor/hydroxi-*cis*-pinonic acid/water double interfacial system at 200 K and 300 K. Simulated free energy profiles are used to map the water uptake mechanism and are separated into energetic and entropic contributions to highlight its main thermodynamic driving forces. Atmospheric implications are discussed in terms of gas/particle partitioning, intraparticle water redistribution timescales, and water vapor equilibrium saturation ratios. Our simulations reveal a strongly temperature-dependent water uptake mechanism, whose most prominent features are determined by local extrema in conformational and orientational entropies near the organic/water interface. This results in a low core uptake coefficient ($k_{o/w}$=0.03) and a concentration gradient of water in the organic shell at the higher temperature, while their effect is negligible at 200 K due to the association entropic term reduction in the free energy profiles. The concentration gradient, which results from non-ideal mixing - and a major factor in increasing LLPS CCN activity - is responsible for maintaining LLPS and low surface tension even at very high relative humidities, thus reducing critical supersaturations. Thermodynamic driving forces are rationalized to be generalizable across different compositions. The conditions under which single uptake coefficients can be used to to describe growth kinetics as a function of temperature in LLPS particles are described.

## 1 Introduction

Aerosol-cloud interactions constitute one of the most important sources of uncertainty in assessments of anthropogenic climate change (IPCC). The number, size and composition characteristics of aerosol influence the number of droplets that can form in cloudy updrafts, which in turn affect the microphysical evolution and radiative properties of clouds and affect climate. Cloud droplets form upon a subset of aerosol, called cloud condensation nuclei (CCN), that become unstable and experience unconstrained growth in the supersaturated water vapor that develops in a cloudy updraft. The dynamics of water uptake on CCN is a critical process that influences the level of supersaturation that can develop in clouds (Raatikainen et al., 2013) and is

affected by the interplay of gas-to-particle transport, interfacial mass transfer and diffusion in the particle phase. Water uptake by particles is controlled by the gas-to-particle transport timescale when the mean free path of gas phase molecules is smaller than the particle size. Particle phase diffusion dominates in glassy and semisolid aerosol, while interfacial mass transfer can be important when the particle size is comparable to or smaller than the gas-phase mean free path.

Limitations on particle growth by interfacial mass transfer can be expressed in terms of the mass accommodation coefficient ($\alpha$), defined as the ratio of molecules absorbed by the particle over the total number of molecules colliding with the surface. $\alpha \sim 1$ for particles whose surface is composed of dominantly water (Clement et al., 1996; Morita et al., 2004; Voigtländer et al., 2007), while $\alpha$ can be very reduced for hydrophobic surfaces. For hydrophobic organic films (e.g.: long chained alcohol) model films $\alpha$ was found to be as low as $10^{-3} - 10^{-5}$ from molecular dynamics (Takahama and Russell, 2011; Ergin and Takahama, 2016; Johansson et al., 2020) and experimental studies (Diveky et al., 2019; Johansson et al., 2020). Gas-to-particle partitioning on secondary organic aerosol (SOA), which constitutes a significant fraction of the total particulate matter on a global scale (Fuzzi et al., 2006), is usually characterized by $\alpha > 0.1$ (Raatikainen et al., 2013; Julin et al., 2014; Krechmer et al., 2017). Modeling studies have long shown that reduced mass accommodation coefficients result in increased droplet number concentration in clouds, owing to the slow condensation of water vapor in the initial stages of cloud formation, which in turn elevates supersaturation and allows for more CCN to activate. For these kinetic delays to be important for climate simulations, the uptake coefficient needs to be less than 0.1 (Raatikainen et al., 2013). To date, although low values of $\alpha$ have been reported for select systems, analysis of droplet formation upon aerosol from a broad range of environments - even aerosol with large amounts of hydrophobic material (Moore et al., 2012; Raatikainen et al., 2013) has not indicated uptake coefficients below 0.1. $\alpha$ also depends on temperature in a system-specific way (Li et al., 2001; Davidovits et al., 2004; Zientara et al., 2008; Davies et al., 2013; Roy et al., 2020). This is typically not accounted for by parcel models of cloudy updrafts, which tend use a single $\alpha$ value to simulate the entire range on which the temperature of a rising air parcel varies (Morales Betancourt and Nenes, 2014), which can lead to systematic errors, whose magnitude may depend on elevation (i.e., as temperature drops).

While the single parameter description, which uses only the mass accommodation coefficient to describe gas-to-particle partitioning in atmospheric systems, is established, in reality the definition of $\alpha$ is somewhat unambiguous. A clear definition of the reference state – i.e., the state beyond which a water molecule is considered to have partitioned to the particle phase – is unclear. The most commonly accepted definition of $\alpha$ uses adsorption at the particle surface or absorption by the first few molecular layers of the particle as a reference state, while a penetration depth-dependent definition (Shiraiwa and Pöschl, 2020) has also been proposed. The latter accounts for particle phase diffusion using an effective accommodation coefficient. This approach turns out to be sufficiently complex to describe water uptake by well mixed particles.

Typical values of $\alpha > 0.1$ observed for SOA CCN suggest that they do not substantially exhibit kinetic delays in gas-to-particle partitioning. Nevertheless, increased CCN activity compared to the one predicted by $\kappa$-Köhler theory has been reported for SOA in particular for liquid-liquid phase separated (LLPS) particles (Prenni et al., 2007; Pajunoja et al., 2015; Liu et al., 2018). LLPS tends to occur in deliquesced SOA-rich aerosolif the O:C ratio of the organics is below 0.8 (Song et al., 2012, 2017; Renbaum-Wolff et al., 2016). Particles formed by LLPS have either core-shell or partially-engulfed morphopolies (Song et al., 2012; You et al., 2014; Gorkowski et al., 2020). LLPS organic shells are non-ideal mixed multilayers, potentially affecting

interfacial mass transfer kinetics in a complex way which may possibly not parameterized by a single mass accommodation coefficient (Krieger et al., 2012; Davies et al., 2013). Increased CCN activity of LLPS particles may arise from i) reduced surface tension due to the high concentrations of organic molecules at the surface of LLPS particles; ii) surface adsorption (Pajunoja et al., 2015; Sareen et al., 2013) or iii) non-ideal mixing. Surface tension reduction is proven in laboratory (Song et al., 2012, 2017; Renbaum-Wolff et al., 2016) and field experiments (Facchini et al., 1999), and its role in enhancing CCN activity has been extensively discussed (Ruehl et al., 2012, 2016; Noziere, 2016; Ovadnevaite et al., 2017). The existence of a long-lived surface-adsorbed states is unlikely when the outer phase has randomly oriented hydrophobic groups, due to the rapid formation of H-bonds seen in MD simulations (Johansson et al., 2020), and was recently experimentally shown to be insignificant (Liu et al., 2018). Studies also have also shown the importance of non-Raoult-type non-ideality by introducing a Flory-Huggins type of representation of the non-ideal Gibbs free energy of mixing in a $\kappa$-Köhler model (Petters et al., 2006; Liu et al., 2018). However, Flory-Huggins theory cannot explain how such non-idealities can enhance CCN activity. Molecular scale understanding of the manifestation of non-idealities in the particle structure can help understan how reduced surface tension and non-ideal mixing effects are linked.

Both the ambiguities surrounding the definition of gas-to-particle partitioning in particles that are not well mixed, and the lack of detailed understanding of the effect of LLPS on cloud droplet growth and activation are closely tied to the limited molecular-scale knowledge about the partitioning process, which has been mapped for only a handful of systems (Sakaguchi and Morita, 2012; Ergin and Takahama, 2016). Nanosecond timescales associated with interfacial mass transfer of a single water molecule to an aerosol particle (Bzdek and Reid, 2017) are conveniently studied by molecular dynamics (MD) simulations, using equilibrium (Bahadur and Russell, 2008), direct impinging (Vieceli et al., 2005; Takahama and Russell, 2011; Johansson et al., 2020) or umbrella-sampling techniques (Sakaguchi and Morita, 2012; Ergin and Takahama, 2016). They have estimated near-unity surface accommodation coefficients on pure water surfaces (Morita et al., 2004; Vieceli et al., 2005; Takahama and Russell, 2011) and reproduced reduced coefficients characteristic of hydrophobic surfaces (Sakaguchi and Morita, 2012; Ergin and Takahama, 2016; Miles et al., 2016; Johansson et al., 2020) as well as size dependence of uptake coefficient in salt nanoparticles (Bahadur and Russell, 2008). In direct impinging simulations, gas-phase molecules are launched with an initial velocity towards the bulk phase in several trials and the mass accommodation coefficient is simply estimated as the ratio of successful trials. This method requires additional techniques to identify the eventual fate of molecules that are adsorbed at the surface but desorbed later without entering the bulk liquid phase. Umbrella-sampling (Torrie and Valleau, 1977) simulations are used to reconstruct the free energy profile of water uptake from a series equilibrium simulations with the position of the gas phase molecule at different distances from the bulk, which is then converted to mass accommodation coefficient using the transition state theory (Sakaguchi and Morita, 2012). In its traditional formulation, umbrella-sampling used to study transfer through liquid surfaces artificially smoothes the effect thermal surface fluctuations due to the way in which averaging is performed (Darvas et al., 2013). A very recent study used well-tempered metadynamics to estimate equilibrium partitioning coefficients of volatile organics (von Domaros et al., 2020). In this work, an alternative approach based on steered molecular dynamics (Steered MD) (Park and Schulten, 2004) is proposed. It has been successful at exploring free energy profiles using out of equilibrium simulations along distance-related reaction coordinates in biophysical contexts (Allen et al., 2014, e.g.).

This paper presents a molecular simulation study aimed at revealing the mechanism of water uptake in a vapor/hydroxy-*cis*-pinonic acid/water double interfacial system at two temperatures, characteristic of the planetary boundary layer and the upper troposphere. Free energy profiles of water uptake are generated using steered MD simulations, and are used to describe the temperature dependence of the water uptake mechanism. Thermodynamic driving forces are identified by decomposing the free energy profiles into entropic and energetic contributions, and the causality between exact molecular scale representation

of non-ideal mixing thermodynamics and observed increased CCN activities is inferred from the thermodynamic description. Knowing the molecular-scale mechanism of water uptake, the usability of a single mass accommodation coefficient to describe gas to particle partitioning is assessed by modeling characteristic timescales and concentration distributions in the organic shell. Finally, different scenarios through which LLPS affects cloud droplet activation and growth are identified and tested in the framework of Köhler-theory.

## 2  Methods

### 2.1  Technical background

The molecular scale mechanism of water uptake is studied through the analysis of free energy profiles. This approach provides a comprehensive overview of the gas-to-particle partitioning process. Molecular simulation methods to calculate free energy profiles, among them umbrella sampling and steered molecular dynamics, rely on forcing the system to follow a pathway along

an aptly chosen set of reaction coordinates that provide a low-dimensional representation of the physical process. In umbrella sampling (Torrie and Valleau, 1977), the reaction coordinate space is mapped on a set of consecutive quasi-equilibrium simulations, with the reaction coordinates restrained at a different value in each simulation. Quasi-equilibrium methods (Sakaguchi and Morita, 2012) provide a free energy estimate that artificially averages out effects of surface fluctuations (as a result of the method used to average the force as a function of the reaction coordinate) (Darvas et al., 2013; Braga et al., 2016; Klug et al.,

2018) whose spatial and temporal scales are similar to those of a typical atmospheric water uptake event.

In steered MD (Park and Schulten, 2004), the system is pulled along the reaction coordinate space with the help of an external harmonic bias at a constant finite velocity or with a constant finite force in several parallel realisations. Jarzynski's equality (Jarzynski, 1997) is used to estimate the free energy from the work profiles collected in each non-equilibrium simulation:

$$e^{-\beta \Delta G} = \langle e^{-\beta W} \rangle, \tag{1}$$

where $\beta = 1/k_B T$ and $k_B$ is the Boltzmann constant and $T$ is the temperature. A formal proof of Jarzinsky's equality for finite size systems, coupled to an external heat reservoir (a typical model of a canonical MD simulation) can be found elsewhere (Cuendet, 2006; Schöll-Paschinger and Dellago, 2006). Besides free energy differences, steered MD can also be used to reconstruct free energy profiles using different reweighting schemes (Gore et al., 2003). The reweighting is necessary as every "pulling simulation" relies on attaching a harmonic spring to the molecular system in the direction of the reaction coordinate

$(s(\mathbf{x}, t))$. This signifies adding an external time dependent bias $(V[s(\mathbf{x}, t)])$ to the Hamiltonian $(H_0(\mathbf{x}, t))$ of the systems whose coordinates are denoted by $\mathbf{x}$, yielding a perturbed Hamiltonian $(H(\mathbf{x}, t) = H_0(\mathbf{x}, t) + V[s(\mathbf{x}, t)])$, which does not correspond

to the unbiased statistical mechanical ensemble in question (Tuckerman, 2010). In this work we use the scheme introduced by Hummer and Szabo (Hummer and Szabo, 2001):

$$e^{-\beta G(s)} = \frac{\sum_t \frac{\langle \delta[s-s(t)]e^{-\beta W_t})\rangle}{\langle e^{-\beta W_t}\rangle}}{\sum_t \frac{e^{-\beta V(s,t)}}{\langle e^{\beta W_t}\rangle}}, \tag{2}$$

where $\langle ... \rangle$ denote averaging over parallel realizations.

According to the second law of thermodynamics, it is only possible to provide an upper estimate of the equilibrium free energy related to a process from the work performed, and the equality between work and free energy holds only in the idealised case of transitions that occur at vanishingly-small velocity. Equations based on Jarzinksy's equality (Gore et al., 2003) can infer free energy profiles from work profiles accompanying finite velocity transition. Jarzinsky's equality ensures that while 135   the individual trajectories, from which the work along the time dependent reaction coordinate ($s = s(t)$) is estimated, drive the system out of equilibrium, the free energy difference is calculated from ensemble averages over the microstates that describe a thermodynamic states along the path (Tuckerman, 2010). Thus instead of a single ideal thermodynamically reversible path, the free energy is estimated from a large number of irreversible pathways (that mimic the realistic uptake process) which are weighted according to their distance from the idealised path in the free energy representation. Jarzinsky's equality yields an 140   estimate of the equilibrium free energy regardless of the velocity of the transition albeit based on a physically realistic set of sample processes.

In practice, Jarzinsky's equality and steered molecular dynamics have been successful at examining finite speed transitions between two states of a molecular system, e.g.: protein or nucleic acid unfolding (Park and Schulten, 2004; Gore et al., 2003). The timescale associated with a single water uptake event (impinging plus transport from the surface to the bulk) is $\sim 10$ ns, 145   surface residence times are $\sim 1$ ns (Bzdek and Reid, 2017). These timescales are matched by the length of individual realizations in our steered MD simulations. Additionally, initializing each realization (out-of-equilibrium simulation) from different starting configurations (Section 2.2) allows impact to happen at any point of the surface, thus incorporating thermodynamically suboptimal pathways - which have finite probability in experiments - are well represented in the sample. This approach confers an advantage for estimating the free energy profile compared to quasi-equilibrium approaches (i.e., umbrella sampling) 150   in which the free energy represents the minimum energy pathway due to the time permitted for sampling the surface. In our systems, finite velocity pulling allows to sample transport through humps and wells of the corrugated intrinsic interface (Section 3.1.1). These corrugations represent thermodynamically different environments (Bartók-Pártay et al., 2008; Darvas et al., 2010a) for contact formation upon impact, and hence different transport pathways which can be relatively far away from the equilibrium one, but occur in real events with a finite probability. Note that exploring suboptimal pathways of interfacial mass 155   transfer can be even more important in modeling transport through multicomponent surface films with heterogeneous lateral distributions.

The steered MD method is illustrated in Figure 1. A sample steered MD trajectory is available as video supplement (https://zenodo.org/record/4902870#.YLsZwS0Rq_U).

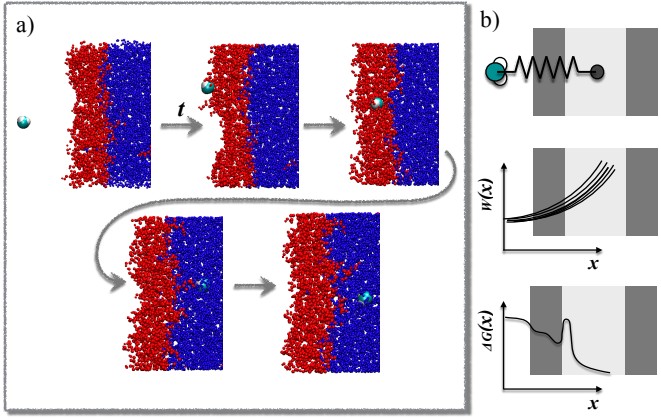

**Figure 1.** a) Example of the evolution of a steered MD simulation. b) Schematic summary of the free energy reconstruction protocol.

## 2.2 Simulation details

Steered MD simulations were performed using the GROMACS 5.1.3 program package (Abraham et al., 2015). The PLUMED 2.5 plugin (Tribello et al., 2014); was used to implement and control the steered MD simulations. The system consisted of a rectangular slab containing 5000 water molecules enclosed between two multilayers of hydroxy cis-pinonic acid (h-CPA), each of 125 molecules. The liquid slab was surrounded by a vapor phase from both sides (Figure 1). The average widths of the layers along the Z axis of the simulation box being $\sim 6.5$ nm, $\sim 2.5$ nm and $\sim 6$ nm for water, CPA and the vapor phase, respectively. The protocol to create and equilibrate such interfacial systems is described elsewhere (Hantal et al., 2010; Darvas et al., 2011b). A single water molecule is placed in the vapor phase of the preequilibrated interfacial system and pulled towards the middle of the aqueous phase along the reaction coordinate ($s(\mathbf{x},t)$) - defined as the interface-normal (Z) component of the distance connecting its center of mass to that of the aqueous phase by harmonic bias having a force constant of k=1000 kJ mol$^{-1}$ nm$^{-1}$. This condition satisfies the stiff spring approximation which enables the system to closely follow the path of the reaction coordinate by becoming the dominant force in the total dynamics Hummer and Szabo (2001), but results in a motion slow enough to mimic diffusive impact. The bias on the reaction coordinate does not constrain the displacement of the pulled molecule in the directions parallel to the macroscopic plane of the interface, thus the impact angle - i.e. the angle between the macroscopic plane of the interface and the vector defined by the initial position of the molecule and its position at the moment of impact - is not constrained to be perpendicular to the surface. This conditions allows to very closely model realistic atmospheric scenarios in which the impact angle is variable and random. Sample impact angles are shown in Appendix A as snapshot series extracted randomly from a three sample simulations.

82 parallel 6 ns-long realizations are performed at two temperatures, 200 and 300 K on the NVT (canonical) ensemble. This number of simulations is sufficient to provide statistically accurate estimates of the free energy profiles. Both the standard deviation and the 95% confidence intervals drop below $0.2k_B$T when the number realizations exceeds 75 (Detailed analysis of

the statistical accuracy of the simulations is shown in Appendix C). The length of the simulation and the distance covered in the direction of the reaction coordinate results in an average pulling velocity of $1.2\,\mathrm{nm\,ns^{-1}}$. The realizations differ in the initial position of the gas phase water molecule in the X,Y plane parallel to the surface of the liquid slab, and in the random seed which is used to set initial velocity distributions. The choice of pulling velocity is a crucial point in setting up the simulation. It should be low enough to ensure that final and initial states represent the equilibrium distribution (Tuckerman, 2010), and also to yield numerically treatable forces. On the other hand, it has to be large enough to be drive the dynamics of the system, and too low transition velocities will undersample higher work pathways. Temperature is kept constant by means of the V-rescale thermostat (Bussi et al., 2007). Water molecules are described by the TIP4P water model (Jorgensen et al., 1983) and h-CPA molecules by the OPLS potential (Jorgensen and Tirado-Rives, 1988). Alkyl groups are treated as united atoms, while other hydrogens are treated explicitly. Long range electrostatics are accounted for by the particle mesh Ewald method (Essmann et al., 1995) beyond a cutoff of 1 nm, while Lennard-Jones interactions are smoothly truncated to zero beyond the same cutoff.

## 3  Results and discussion

Free energy profiles are interpreted in terms of the local characteristics of the simulated systems and a detailed mechanism of water uptake is proposed. The decomposition of the free energy profiles into enthalpic and entropic contributions allows us to identify the thermodynamic drivers of interfacial mass transfer of water at the two studied temperatures. The effect of the complex water uptake mechanism on particle growth and activation is then assessed by estimating interfacial transfer coefficients. Finally, implications for droplet formation are discussed in the light of partitioning timescale estimates, intraparticle distribution of water within the organic shell, and equilibrium saturation ratios.

### 3.1  Structural and thermodynamic characteristics

#### 3.1.1  Intrinsic density profiles

Local nanoscale fluctuations of fluid interfaces from capillary waves create surface corrugations (Rowlinson, 1982) whose amplitude varies between 0.3 and 1 nm depending on the chemical composition of the surface (Hantal et al., 2010; Darvas et al., 2011a). These corrugations do not average out during typical timescales of interfacial mass transfer of a water molecule and are enhanced when a molecule crosses the interface (Benjamin, 1993; Karnes and Benjamin, 2016; Benjamin, 2019). Thermal fluctuations are not accounted for by traditional surface definitions and cause artificial smoothing of the surface and systematic errors in the selection of surfaces molecules (Bartók-Pártay et al., 2008; Jorge et al., 2010). Intrinsic surface analysis – i.e. decoupling surface fluctuations from the interfacial properties (Appendix A) – of molecular simulation trajectories is necessary to resolve the effect of capillary waves on density (Jorge et al., 2010) and free energy profiles (Darvas et al., 2013; Braga et al., 2016; Klug et al., 2018). Results are sharper and contain features which are smoothed out by fluctuations using traditional surface definitions. The intrinsic surface analysis thus reveals any possible layering near the interface -if the layered structure exists - that is otherwise smoothed out by the smearing effect of the capillary waves. Layering is typically weak in

aqueous phases near an organic counterphase. Earlier work reports one pronounced layer and a hardly noticeable second layer for instance in water/dichloroethane systems(Jorge et al., 2010).

In this work, intrinsic analysis is limited to the calculation of density profiles anchored to the fluctuating surface by the ITIM algorithm (Sega et al., 2018), which allows to establish qualitative connections between the free energy profiles and the local perturbations of solution structure and thermodynamic properties in the vicinity of interfaces. The algorithm used in this work has the advantage that instead of defining the covering surface it selects the molecules of the interface in an exact manner, and thus allows for calculating a structural and collective properties selectively for the true interface. Repeating this analysis several time consecutively provides sequential molecular layers that can be analysed separately. This advantage is used for the calculation of orientational distribution of waters at the surface.

Shaded blue and yellow areas in Figure 2 a) and b) show the intrinsic mass density profiles of the aqueous and the organic phase. Organic density profiles reveal that the thickness of the organic phase is $\sim 1.8$ and $\sim 2.1$ nm at 200 and 300 K, respectively, measured at 5% of the average height of the profile. It is sufficient to accommodate a disordered double layer, but is too low to allow the formation of a bulk phase. The lack of a bulk phase is evidenced by the two neighboring peaks – characteristic of interfaces – in the density profile of the organic phase with no plateau at the bulk phase density ($1.2 \, \mathrm{g \, cm^{-3}}$) in between. The layer by layer density profiles obtained from the ITIM algorithm lead to the same conclusion (Appendix A). A $\sim 85\%$ local drop in the total density (sum of water an organic density) compared to that of bulk aqueous phase can be observed at the Gibbs dividing surface of the two condensed phases at both temperatures. The affected region is wider at 300 K. The interface in our systems is mixed and the density drop results from the presence of voids larger than the typical distances between the molecules in the bulk. A more detailed analysis of the interface structure and a demonstration of the voids is found in Appendix A. The interfacial region is characterized by an overlap between the organic and the water density profiles which is due to partial miscibility of the two phases. The width of this mixed region is 1.5 nm at 300 K and 1.2 nm at 200 K disregarding the small peak of organics in the bulk aqueous phase. The aqueous phase has two consecutive peaks characteristic of the first two molecular layers, followed by a bulk phase plateau. The second peak is less pronounced than the first and the shape of the profile is consistent with previous results(Bresme et al., 2008; Jorge et al., 2010). The enhanced structure in the intrinsic density profile of water compared to the global profiles is shown in Appendix A. The interface and subsurface structure are similar, but more pronounced at 200 K as result of reduced thermal motion which preserves layered structure at low temperatures.

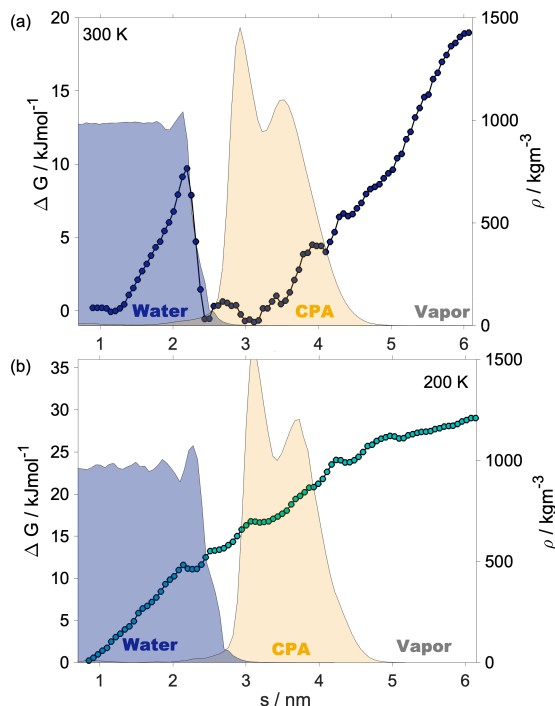

**Figure 2.** a) Free energy and density profiles at T=300 K; b) the same profiles at T=200 K

### 3.1.2 Free energy profiles

Helmholtz free energy profiles, which represent the thermodynamic state function on the canonical ensemble are calculated. While the Helmholtz free energy is formally different from the experimentally determined Gibbs free energy, in systems containing incompressible condensed phases the additional $p\Delta V$ term is negligibly small; thus the two quantities can be assumed to be the same within a small error. Free energy profiles with the reference state assigned to the bulk aqueous phase are overlapped with the intrinsic mass density profiles in Figure 2 a) and b). Profiles are significantly different at the two temperatures which suggests that mechanism of gas-to-particle partitioning in LLPS particles is temperature dependent. Despite differences, simulations show two common characteristics: i) the negative free energy difference ($\sim -19.5$ kJ mol$^{-1}$ at 300 K and $\sim -29.0$ kJ mol$^{-1}$ at 200 K) between the bulk aqueous and the vapor phase, which underlines that overall water uptake from the vapor phase is thermodynamically favored, and, ii) the lack of a free energy barrier at the vapor/organic interface, which is the main difference between free energy profiles of transfer through hydrophobic media, for which large maxima have been observed leading to reduced surface accommodation coefficients (Sakaguchi and Morita, 2012; Ergin and Takahama, 2016). The free energy differences between the liquid and the vapor phase agree within 25% and 9% with the data estimated from experimental densities and vapor pressures at 300 and 200 K respectively. Temperature dependent trends are consistent with expectations, however, the simulations only reproduce experimental thermodynamic data semi quantitatively, which can

be attributed to the accuracy of transferable potentials combined with complexity of the studied system. The detailed description of the comparison can be found in Appendix C.

The free energy profile at 300 K begins with a plateau characteristic of the vapor phase, followed by a steep decrease ($\sim 5$ kJ mol$^{-1}$) at the vapor/organic interface. A monotonic decrease in the free energy profile characterizes the transfer of the water molecule in the organic phase. The lack of a plateau is related to the fact that the organic layer is not thick enough to accommodate a bulk phase, which results in a position dependent anisotropy caused by the proximity of both interfaces in combination with the partial dissolution of the water in the organic phase, evidenced by non-zero water density up to s$\approx 3.5$ nm. The presence of dissolved water molecules results in the near 0 free energies in the inner half of the organic phase, and a small maximum (0.6 kJ mol$^{-1}$), considerably smaller than the energy of thermal motion ($3/2k_BT$), which is thus considered as statistically insignificant, and which can however be potentially associated with the increased order in the first molecular layer in the organic phase. The free energy profile has a local minimum at the organic/water interface, and a maximum ($\sim 9.8$ kJ mol$^{-1}$) corresponding to the first molecular layer of water, seen as a peak in the intrinsic density profile, beyond which the profile smoothly decreases to the reference state value in the bulk aqueous phase. The free energy profile at 200 K is considerably smoother; the magnitude of none of the features exceeds the energy of thermal motion ($3/2k_BT$) at the corresponding temperature, thus any small local minimum or maximum can be viewed as statistically insignificant. The minimum and the peak at the organic/water interface characteristic of the room temperature profile cannot be observed at low temperature. While vapor-to-organic transfer is a favorable and barrierless transition at both temperatures, the transport between the organic shell and the core is hindered by the presence of the free energy barrier at 300 K, which diminishes at low temperature. We acknowledge that the profiles assume somewhat unusual shape in the region where they decay to the bulk aqueous phase value, and attribute it to the large mutual solubility of the two phases which results in an extended region where both water and h-CPA molecules are found at varying concentration. Large temperature induced differences suggest that the thermodynamic driving forces are strongly temperature dependent. One possible explanation is that they are of entropic natrure, and their strength is explicitly scaled by the temperature through the $-T\Delta S$ term.

### 3.1.3 Internal energy profiles and entropy

To test the hypothesis that features of the free energy profile are of entropic origin, internal energy profiles are calculated and shown in Figure 3. The sum of the interaction energies of the pulled water with the surrounding molecules is used as a surrogate for the internal energy, the calculation is described in details in Appendix B. In principle, the total entropy profile can be obtained trivially by subtracting the internal energy profile from the free energy profile, however given that neither the free energy profiles, nor the internal energy profiles can be assumed to be quantitative, the physical significance of their difference is questionable, hence total entropy profiles are not shown here. Instead, distinct entropy contributions are analysed in detail, which have been selected to explain the features of the free energy profiles.

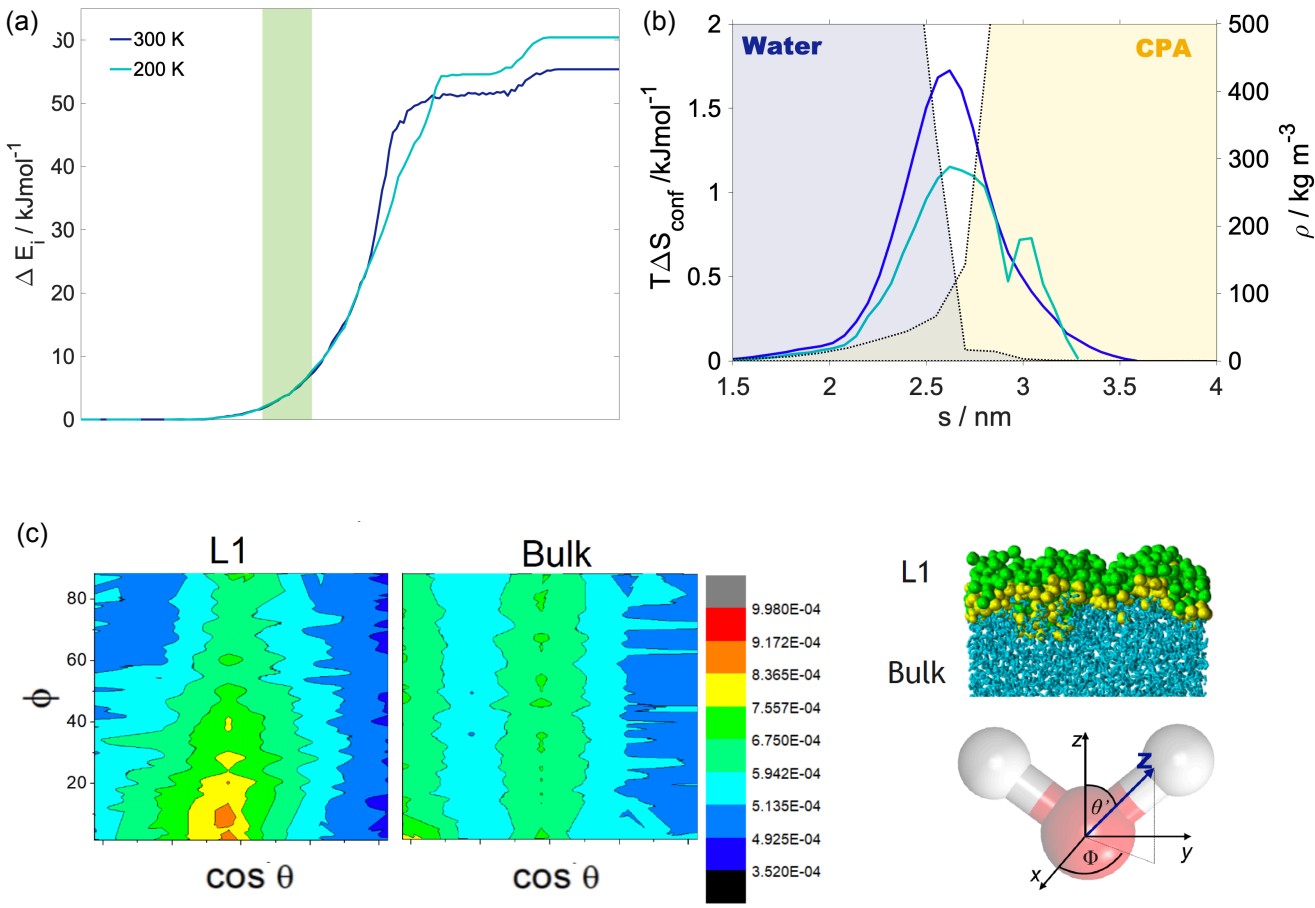

**Figure 3.** a) Internal energy profiles. b) The conformational entropy in the interfacial region overlapped with intrinsic density profiles of the aqueous (blue shaded) and the organic (yellow shaded) phase. c) Left: Orientational maps of the water molecules in the first molecular layer (L1) of the aqueous phase and in a randomly selected layer in the bulk phase. Right: An equilibrium snapshot from a simulation showing the first two molecular layers (green and yellow balls) and the bulk phase (cyan sticks) as determined by the ITIM algorithm; the definition of the orientational vectors of the water molecule.

Internal energy profiles in Figure 3 a) show that water uptake is energetically favorable, the difference between the pulled water molecule being in the vapor phase and in the aqueous phase is $\sim -55$ kJ mol$^{-1}$ at 300 K and $\sim -60$ kJ mol$^{-1}$ at 200 K. Internal energy differences correspond approximately to the formation of 3-4 hydrogen bonds by the pulled molecule if the average energy of a hydrogen bond is assumed to be 15-20 kJ mol$^{-1}$ (Wendler et al., 2010), which is also supported by the hydrogen bond profile calculated for a randomly selected realization (Appendix A). Internal energy profiles are smooth and show a similar overall behavior at the two temperatures. A small drop in the internal energy whose magnitude is $\sim 3$ kJ mol$^{-1}$ and $\sim 5$ kJ mol$^{-1}$ at the 200 and 300 K respectively indicates the formation of contact with the organic phase. It is however not sufficiently low compared to the internal energy observed in the subsequent bulk organic phase to energetically stabilise a

surface adsorbed state. Both internal energy profiles have a plateau spanning the outer half of the organic phase, where water molecules from the bulk aqueous phase do not penetrate. This is followed by a smoothly decreasing part which corresponds to the pulled molecule forming an increasing number of hydrogen bonds with both h-CPA and the water molecules dissolved in the organic phase. Near the organic-water interface and in the aqueous phase the internal energy profiles are close to identical at the two temperature, suggesting that specific features of the free energy profile are of entropic origin.

Configurational entropy profiles calculated using Schlitter's formula (Baron et al., 2006) as explained in Appendix B are close to constant throughout the condensed phase, $-T\Delta S_{\text{config}} = 25$ kJ mol$^{-1}$ at 300 K and 14 kJ mol$^{-1}$ at 200 K. They are thus not responsible for any of the features in the free energy profiles. Interfacial entropy ($T\Delta S_{IF}$) at the vapor/organic interface is approximately 5 kJ mol$^{-1}$ at 300 K and is negligibly small at 200 K (Appendix B). Despite of the positive value of interfacial entropy the overall entropy difference between the vapor and the organic phase is a small negative value, which — together with the moderate change in enthalpy at the interface — is responsible for the lack of surface adsorbed states. A possible explanation for the overall negative entropy is the ordering of the organic molecules at the surface which reduces orientational degrees of freedom in a similar manner as described later for the water/organic interface.

Conformational entropy profiles, calculated from the mole fraction profiles of the water and the organic molecules (Appendix B), are shown in Figure 3 c). They exhibit a peak located at the organic/water interface at both temperatures and are close to zero elsewhere. These peaks represent the effect of non-ideal local mixing of two phases in this region. The conformational entropy peak is slightly higher at 300 K, mostly due to the higher temperature. This and the local $\sim 85\%$ decrease in the total density in this region explain the presence of the free energy minimum at the organic/water interface. Lower density ensures the reduction of steric hindrance for any conformations without loosing hydrogen bonds. These explain the minimum in the free energy profile observed at the water/organic interface at 300 K. High conformational degrees of freedom manifest in subsequent detaching/attaching of the pulled molecule between the organic and aqueous phase in varying orientations, which can be observed only in the 300 K simulations (video supplement available). The conformational entropy profile peak in the interfacial region is significantly ($\sim 40\%$) smaller at 200 K. Additionally, the low density region is narrower due to the pronounced shoulder in the water density profile (Figure 2), thus steric effects are not as effectively reduced as at 300 K. The joint decrease of the magnitude of both contributions leads to the disappearance of the free energy minimum at the organic/water interface at 200 K.

Increased orientational order of molecules at interfaces of molecular liquids locally reduces orientational entropy, which is responsible for the free energy peak coinciding with the first two molecular layers of water in the 300 K simulations. Interfacial molecules adapt preferred orientations owing to topological anisotropy of intermolecular interactions, in contrast to the bulk where the close to isotropic energetic environment ensures nearly random orientations. To illustrate the differences between the orientational preferences of interfacial and bulk water molecules we calculate the joint distribution of the two angles defined in the bottom right panel of Figure 3 d) in the first molecular layer and in a randomly selected molecular layer from the bulk aqueous phase. Details of the calculation are summarized in Appendix B. The two dimensional joint distribution of these two angles (Figure 3 d), top panel) is able to define the orientation of a rigid body having a $C_{2v}$ point group symmetry with respect to an external axis. The peak shows that in the first layer water molecules tend to lie parallel to the surface, whereas the bulk

phase distribution shows no significant preferences. Increased orientational order results in a decrease in orientational entropy, corresponding to the maximum of the free energy profile. The second layer of water has similar orientational preferences (Appendix B), which explains the shoulder of the free energy profile near the second molecular layer. Similar results are found for 200 K (Appendix B), thus the observed temperature dependence results again from the explicit scaling of the importance of entropy with temperature. Orientational entropies are also estimated in 1 dimension (Appendix B), but due to symmetry reasons they do not give a quantitative description of the ensemble of ordering effects.

### 3.1.4 Generalization of the driving forces

The question whether the thermodynamic driving forces identified in the previous section are system specific or generalizable over a wider range of compositions that can yield LLPS particles is important for determining the level of confidence with which implications to atmospheric aerosol can be stated based on these simulations. The most prominent characteristics of the free energy profiles are i) the lack of a minimum at the vapor/organic interface, ii) the minimum at the organic/water interface and iii) the maximum corresponding to the first layer of the aqueous phase.

Experimental and molecular simulation studies show that driving forces which lead to the appearance of these features are generally present in interfacial systems involving molecular liquids or solids. i) The value of the free energy at the vapor/organic interface is moderated by the increased order of organic molecules which has been observed from experiments and simulations in pure water (Cipcigan et al., 2015), pure organics (Darvas et al., 2010a) as well as in concentrated (Darvas et al., 2010b) and dilute aqueous organic solutions (Takamuku et al., 1998; Pártay et al., 2008; Pojják et al., 2010; Ghatee et al., 2011; Makowski et al., 2016). ii) The density drop in the interfacial region and the enhanced local mixing which invoke the free energy minimum at the organic/water interface at 300 K are characteristic of any liquid/liquid interface (?Fernandes et al., 1999; Jedlovszky et al., 2002; Hantal et al., 2010; Jorge et al., 2010; Darvas et al., 2011b, 2013). The extent of local mixing is a non trivial function by the hydrophilicity of the organic compound and the types intermolecular interactions. iii) Orientational preferences of water molecules near liquid/liquid or liquid/solid interfaces, which account for the maximum of the free energy profiles are also generally present in interfacial systems. They have been observed in molecular simulations several times at various liquid/liquid interfaces such as carbon tetrachloride/water (Hantal et al., 2010; Kertész et al., 2014) or dichloroethane/water (Hantal et al., 2010) and proved to be enhanced next to a solid counter phase (Kertész et al., 2014).

The strong temperature dependence follows directly from the entropic nature of these driving forces and the definition of the free energy. Consequently, besides the presence of the driving forces, their temperature dependence is also expected to be generally valid. LLPS is known to form if the O:C<0.8, the organic components in the examples listed above cover the complete O:C ratio range. Thus our findings are likely not system specific, and represent a typical behavior for sparingly soluble organic compounds present in the atmosphere, although exact values of the thermodynamic quantities can vary from system to system. The following implications are thus quantitatively only valid for the water/h-CPA/vapor system, while qualitatively similar behavior can be expected for a wider variety of LLPS forming cases.

## 3.2 Implications for water uptake and particle growth kinetics

### 3.2.1 Atmospheric relevance of the simulated systems

The simulations presented in this paper aim to provide a qualitative description of how the change in temperature affects the mechanism of water uptake in a particle formed at a lower height (higher temperature) while rising. LLPS particles formed at lower altitudes will likely preserve the phase separated structure also at high altitudes, therefore the same system structure is used at both temperatures.

The first temperature is chosen to represent conditions near the boundary layer or standard laboratory conditions. At 300

K the particles are trivially in the liquid state and the O:C ratio of 0.4 of h-CPA suggests that LLPS is a realistic assumption(Song et al., 2012). 200 K is chosen to model conditions in the upper troposphere. We acknowledge that is an extremely low temperature to examine, however it is common in molecular simulations to use exaggerated conditions in order to accentuate trends along variables such as temperature or pressure, e.g.:(Darvas et al., 2012). As seen in the previous section the structure of the aqueous phase is more pronounced 200 K than at 300 K, however the observed density profiles are very differ-

ent from that of crystalline ice, which is consistent with the difficulty to produce crystalline ice spontaneously from atomistic simulations(Matsumoto et al., 2002).The phase state of the system at 200 K is inferred from the diffusivity of water, which is calculated from 3 randomly selected trajectories using the Einstein equation(Einstein, 1905). The obtained average value of $D{\sim}10^{-13}$ m$^2$s$^{-1}$ suggests our system is in a semi-solid state. The glass transition temperature of pure h-CPA inferred from its molecular weight and O:C ratio (Shiraiwa et al., 2017), is approximately 190-210 K. This together with the low O:C ratio

suggests that similar particle would be phase separated and either glassy or semi-solid. A recent study shows that semi-solid LLPS comprises the domimant phase state of SOA in the upper troposphere(Schmedding et al., 2020). Hence, despite of the exaggerated temperature our system is a valid model of a typical SOA particle in the upper troposphere.

The simulated systems consist of a macroscopically flat model of an a LLPS particle. The flat surface approximation is a generally accepted model of larger particles, where the surface curvature can be neglected. For instance, for aqueous particles

mass accommodation coefficients on the planar surface become equal those estimated for spherical particles beyond particle diameters on the order of 20-30 nm(Barclay and Lukes, 2019). We stress once again, that for reasons described in the Methods section impact angles are not limited to 90°, and are randomly selected by the thermal motion of the particle in the vapor phase, thus our model closely mimics realistic gas-particle collisions.

### 3.2.2 Interfacial transfer coefficients

Interfacial transfer coefficients are estimated from the activation free energies characteristic of the vapor-to-organic ($k_{v/o}$), organic-to-water ($k_{o/w}$) transfer using the transition state theory as:

$$k_{i/j} = \exp\left[\frac{-\Delta_{ij}^{\#}G}{RT}\right],$$ (3)

where $i$ and $j$ indicate the phases forming the interface in question. The corresponding activation free energies are illustrated using the 300 K free energy profile in Figure 4 . While the transfer coefficient calculated here is conceptually different from the mass accommodation coefficient used in atmospheric applications, chemical kinetic frameworks are widely used to estimate $\alpha$ from free energy profiles (Grote and Hynes, 1980; Taylor and Garrett, 1999; Truhlar and Garrett, 2000; Sakaguchi and Morita, 2012; Ergin and Takahama, 2016). We note that due to the lack of a well-defined bulk phase plateau in the organic phase we choose an average free energy characteristic of the middle of the organic phase as a reference value (0.6 kJ mol$^{-1}$). This choice is arbitrary and a different definition may slightly modify obtained transfer coefficients, albeit without significantly altering values, trends and conclusions. Vapor-to-organic (surface) transfer coefficient ($k_{v/o}$) are near unity at both temperatures, whereas organic-to-water (core) transport is characterized by $k_{o/w} = 0.03$ at 300 K, and $k_{o/w} = 1$ at 200 K.

The vapor-to-organic transfer coefficients resemble most closely the commonly used definition of the mass accommodation coefficient, which considers either adsorption at the surface or absorption in the first few molecular layers of the particle phase as the final state of gas-to-particle partitioning. Due to the lack of maxima or minima at the vapor/oragnic interface, in our systems surface adsorbed states of the water are thermodynamically indistinguishable from those with the water absorbed by the subsurface region.

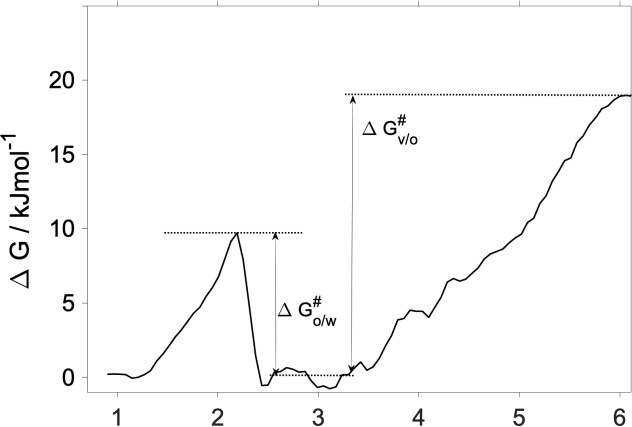

**Figure 4.** a) Definition of activation free energies at the different interfaces at T=300 K.

The organic-to-water transfer coefficient ($k_{o/w}$) is reduced compared to the mass accommodation coefficients ($k_{v/o}$) at room temperature owing to the free energy barrier corresponding to the first molecular layer of water. As opposed to mass accommodation coefficients which show no temperature dependence, $k_{o/w}$ follows a similar trend with temperature as that observed for water uptake on a hexadecanol monolayer (Davies et al., 2013), and on pure water (Davidovits et al., 2004). The fact that core uptake coefficients differ significantly from mass accommodation coefficients highlights the possibility that the traditional representation of water uptake by a single value of $\alpha$ or an effective uptake coefficient may have to be complemented by a temperature-dependent core uptake term to fully describe water uptake by phase-separated aerosol.

### 3.2.3 Mass accommodation coefficients

Mass accommodation coefficients defined by vapor-to-organic transfer coefficients are near unity, and agree well with the globally representative values proposed in modeling studies (Raatikainen et al., 2013), as well as with recent experiments which report large mass accommodation coefficients (Liu et al., 2019) and unhindered gas-to-particle partitioning of water and organics in phase separated particles (Gorkowski et al., 2017). While global datasets of CCN concentrations can usually be described by $0.1 < \alpha < 1$, implying uninhibited water uptake (Raatikainen et al., 2013), compressed hydrophobic organic films can result in mass accommodation coefficients as low as 0.001. In such systems, $\alpha$ is correlated with the integrated carbon number density (Ergin and Takahama, 2016) of the organic layer, estimated as the integral of the density profile along the outer half of the organic layer divided by its width. The integrated carbon number density for our systems is 14 nm$^{-3}$ which for a completely hydrophobic film comprised solely of aliphatic CH moieties would correspond to $\alpha \sim 0.01$ (Ergin and Takahama, 2016) instead of the observed near unity values. In the presence of hydrophilic functional groups — which form hydrogen bonds and dipole-dipole interactions — such a simple descriptor is insufficient to reliably predict mass-accommodation coefficients.

Reduction in mass accommodation coefficients result in increased CCN number concentrations. The explanation for this effect is that droplet number concentrations in ambient clouds depend on the maximum supersaturation, $S_{\text{max}}$, which is largely determined by the condensation rate of water vapor on the growing droplets. Reduced $\alpha$ values correspond to low water vapor condensation rates - which allow supersaturation to develop for longer periods of time (compared to when $\alpha$ is larger), resulting in increased $S_{\text{max}}$ and droplet concentrations. Models simulations predict a mild 1.2-fold increase in CCN number concetrations for $\alpha = 0.1$, a 1.5-1.8-fold increase for $\alpha = 0.01$ and 2-2.5 fold increase for $\alpha = 0.001$ (Raatikainen et al., 2013). In this framework, near-unity mass accommodation coefficients obtained from our simulations at both temperatures are not expected to alter droplet number concentrations, further supporting that kinetic delays do not explain the increased CCN activity of LLPS aerosol.

### 3.2.4 Core uptake coefficients

Hindered mass transfer of water between the organic shell and aqueous core together with uninhibited mass accommodation of water in LLPS aerosol results in different condensational growth rates of the core and shell, leading to a dynamic retention of water by the organic shell. We use the $k_{v/o}/k_{o/w}$ ratio to estimate the extent of dynamic retention of water by the organic phase. The value is approximately unity at 200 K, and about 30 at 300 K. The growth rate of the organic shell is thus substantially larger at 300 K than that of the core, which in agreement with results of multilayer kinetic model (KM-GAP) calculations which also evidence faster condensational growth of the particle shell than of core (Shiraiwa et al., 2013). This suggests that the aqueous core contains less water and the organic shell is more dilute at any time during growth of the particle than predicted assuming that mass transfer kinetics can be described by a single $\alpha$.

Retention of water in the organic shell due to reduced core uptake coefficient (organic-to-water or vapor-to-water transfer coefficient) may affect equilibrium properties (vapor pressure and surface tension) which determine cloud droplet growth and activation. Increased water content of the organic shell can increase vapor pressure and surface tension, which both affect

cloud droplet growth. In the extreme case, inhibited transport between the shell and the core, indicated by reduced core uptake coefficients, may invoke swelling and — depending of the solubility of the organic compounds — dissolution of the shell. These hypotheses can be relevant if the vapor phase is in dynamic equilibrium with the organic shell containing an increased amount of water and this equilibrium is unaffected by the presence of the aqueous core, in other words water uptake happens only by the shell. For this condition to hold gas-to-particle partitioning timescale of water ($\tau_s$) should be significantly shorter than timescale of transfer from the shell to the core ($\tau_c$). Timescales are estimated as the e-folding times of condensation-evaporation, assuming that both gas-to-particle and shell-to-core partitioning can be described as (Saleh et al., 2013):

$$\tau = \frac{1}{2\pi DFdN},$$

(4)

where $F = (1+Kn)/(1+0.3773Kn+1.33Kn(1+Kn)/\alpha)$ is the Fuchs-Sutugin correction factor, with $Kn$ being the Knudsen number and $\alpha$ the transfer coefficient of the given process. For gas-to-particle partitioning $Kn = 10^{-2}, 10^{-1}, 10^0$, $\alpha = 1$, the gas phase diffusion coefficient of water is estimated at 0.26 cm$^2$s$^{-1}$ and 0.128 cm$^2$s$^{-1}$ at 300 and 200 K (Pruppacher and Klett, 2010), while $d$ is the diameter of the particle (100 nm). For core-to-shell partitioning $Kn$ is approximated as the ratio of the diameter of the h-CPA molecules and the width of the organic layer, $\alpha = k_{o/w}$, $d$ is the core diameter, and the diffusion coefficient ($D$) is varied between $10^{-3}$ and $10^{-9}$ cm$^2$s$^{-1}$. The ratio between the core and the full particle diameter is the same as the ratio of the width of the organic shell and the total system width. Assuming the particle number concentration ($N$) to be 1000 cm$^{-3}$, the equilibration timescale of gas-to-particle partitioning $\tau_s$ is on the order of 1-2 minutes, similar to timescales reported for the equilibration of semivolatile molecules(Saleh et al., 2013). $\tau_c/\tau_s > 10^3$ in every case (Figure 5); thus the hypothesis of equilibrium between the organic shell and the vapor phase holds for our system at both temperatures and the water uptake by the particle core is somewhat hindered. The ratio of the timescales is at least three orders of magnitude even when the core and shell uptake coefficients are assumed to be both unity (200 K), owing to the difference between gas and particle phase diffusivity. Reduced core uptake coefficients further increase this difference by orders of magnitude. In summary, as expected the equilibration timescale of gas-to-particle partitioning is sufficiently short to assume equilibrium between the vapor phase and the organic shell, while core uptake may not reach equilibrium within a typical model of cloud updraft.

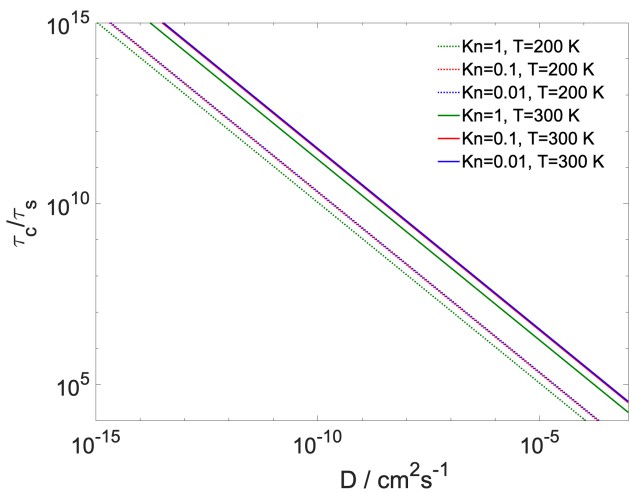

**Figure 5.** The ratio of characteristic timescales of shell-to-core and gas-to-particle partitioning as function of bulk phase diffusivity of the water in the organic shell at varying gas phase Knudsen numbers and temperature. At 300 K model calculations are presented using both the lower and the upper of the core uptake coefficient.

### 3.2.5 The effect of bulk diffusion and non-uniform concentration distribution

The non-negligible free energy difference between the vapor/organic and the organic/water interface potentially alters the above described uniform increase of the water concentration in the shell and carries consequences for droplet growth and activation. A more detailed understanding of these effects can be obtained by converting the free energy profile using the expression $\Delta G(s) = -k_B T \ln c_{\text{rel}}(s)$ into a probabilistic density profile $c_{\text{rel}}(s)$, which corresponds to an equilibrium concentration profile of the condensing water in the organic phase at arbitrary values of the vapor pressure. This equilibrium concentration profile is calculated under the assumption of instantaneous diffusion and bears no information about the surrounding relative humidity (RH), and is valid for a mostly-organic shell (as the free energy profiles were derived for pulling simulations of single water molecules). The effect of non-instantaneous bulk phase diffusion of water and RH are accounted for by a correction factor, $f$:

$$c_{\text{rel}}(s) = \exp\left[\frac{-\Delta G(s)}{k_B T}\right] f(D_p, C_v); \ f(D_p, C_v) = \frac{1}{1 + \frac{\alpha \omega C_v}{4 D_p \rho_p}} \ . \tag{5}$$

$C_v$ and $D_p$ are the vapor phase concentration of water and the bulk phase diffusion coefficient of water in the organic phase, respectively, $\omega$ is the mean thermal velocity of water in the vapor phase, $\alpha$ is the mass accommodation coefficient, and $\rho_p$ is the density of the organic phase (1.2 $\text{g cm}^{-3}$ in our calculation). The correction factor was adapted from Shiraiwa and Pöschl (2020), who derived an expression to account for the effect of diffusivity on gas-to-particle partitioning by introducing a penetration-depth-dependent definition of the mass accommodation coefficients of organic molecules absorbed by aerosol particles used in particle growth kinetic models (Shiraiwa et al., 2012). Our model calculations are performed at five RH values in the range of 50-98%; using 13 different diffusion coefficients between $10^{-15}$ and $10^{-3}$ $\text{cm}^2\text{s}^{-1}$, characteristic of liquid and

semisolid particles. $c_{rel}(s)$ is normalized to the 0-1 range, with 0 and 1 corresponding to the minimum and maximum of the $c_{rel}(s)$ profile within the organic phase at 300 K.

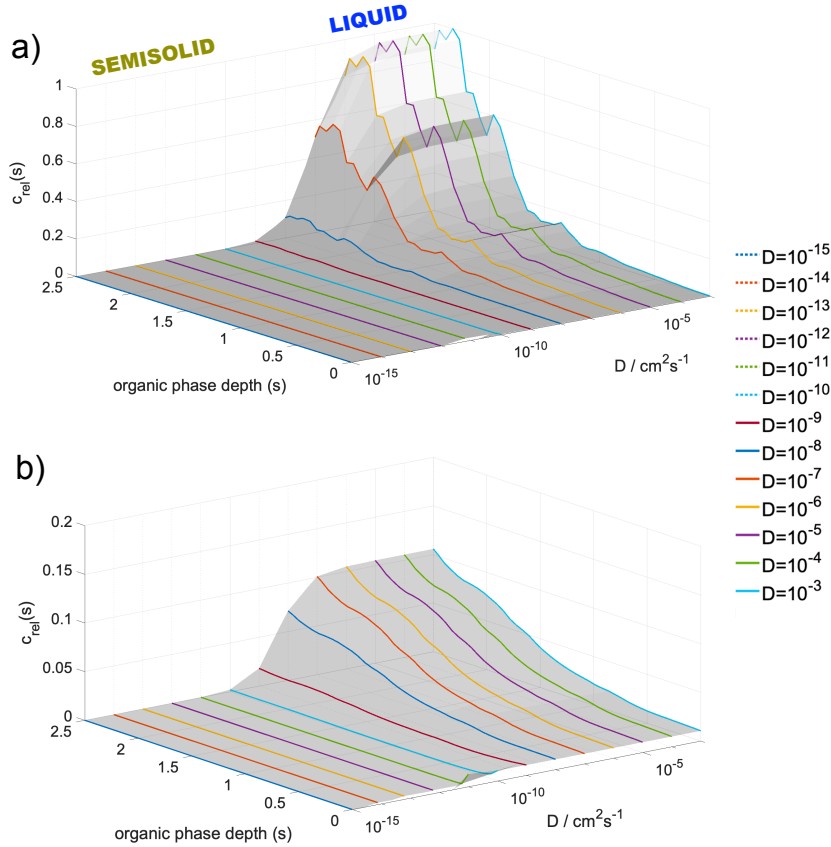

**Figure 6.** Modeled, diffusivity-corrected concentration gradient of water in the organic phase at a) 300 K and b) 200 K at 98% RH and at varying diffusion coefficients. Organic phase depth describes the distance from the vapor/organic interface, zero corresponds the Gibbs dividing surface between the vapor and particle phase

Figure 6 shows the equilibrium diffusion-corrected concentration profiles at RH=95% at 300 K (a) and at 200 K (b). Con-
densing water molecules show a strong preference to be accommodated near the organic/water interface, while the vicinity of the vapor/organic interface is depleted in water at 300 K (Figure 6) a)) for bulk phase diffusion coefficient values characteristic of the liquid phase ($10^{-3} < D_p < 10^{-9}$ cm$^2$s$^{-1}$). A much less pronounced concentration gradient, about an order of magnitude smaller than that found at room temperature, can be observed at 200 K if liquid phase diffusivities are assumed. The effect of diffusion is negligible in the $10^{-3} < D_p < 10^{-6}$ cm$^2$s$^{-1}$ diffusivity range. The steepness of the concentration gradient is
reduced by 40 and 85% for $D_p = 10^{-7}$ and $D_p = 10^{-8}$ cm$^2$s$^{-1}$ at 300 K, and by 30 and 82% for $D_p = 10^{-8}$ and $D_p = 10^{-9}$

cm$^2$s$^{-1}$ at 200 K. Slow diffusion in highly viscous liquid and semisolid states ($D_p < 10^{-10}$ cm$^2$s$^{-1}$) cancels the effect of thermodynamic preferences, and result in uniform concentration profiles at both temperatures. Concentration profiles are not sensitive to varying RH, thus only one characteristic example is shown in Figure 6. Water's diffusion coefficient in h-CPA is estimated from separate unpublished MD simulations to be $10^{-5} - 10^{-6}$ cm$^2$s$^{-1}$ at 300 K depending on concentration, and
$10^{-7} - 10^{-8}$ cm$^2$s$^{-1}$ at the lower temperature, similar values have been also reported based on experiments (Lienhard et al., 2015). This means that room temperature concentration profiles are virtually unaffected by bulk phase diffusion, as liquid-like diffusivities apply. Diffusion control is only probable for diffusion coefficients characteristic in semisolid particles. At 200 K, the originally small concentration gradient is further reduced by slow diffusion. Bulk phase diffusion becomes the governing process for water uptake at low temperatures, regardless of the assumed value of the diffusion coefficient.

The quantitative description is only valid for the system studied and magnitudes may vary as a function of the composition of the organic shell. However, like main driving forces of the water uptake process, the formation of the concentration gradient can be expected in generic LLPS particles regardless of their actual chemical composition. The shape of the concentration profiles may change for a thicker organic layer having a bulk phase corresponding to a constant plateau in the free energy profile, which converts into a constant concentration region in the middle of the organic phase. Nevertheless, the maximum of the
concentration profile coincides with the minimum of the free energy profile at the organic/water interface, which is determined predominantly by local entropy increase due to the lower density and increased conformational degrees of freedom, which is universal at boundaries between condensed phases. Similar considerations are valid for the minimum of the concentration profile, which is observed at the vapor/organic interface, whose value mainly depends on an interplay between intermolecular interactions, orientational order of the vapor/organic interface and interfacial entropy, which are largely insensitive to the
thickness of the organic phase. The significant enrichment of the water/organic interfacial region in water may lead to a local dissolution of the organic phase. However, even when dissolution of the organic phase occurs, the strong preference of water molecules to be accommodated in the inner part of the organic shell results in depleted water concentrations at the surface, which ensures the presence of an organic rich film at the surface, and hence maintains low surface tensions despite of the elevated water concentration in the organic shell even when relative humidity approaches 100%. This is consistent with recent
experiments and model calculations which conclude that LLPS persists up to very high relative humidities (Liu et al., 2018).

### 3.2.6 Equilibrium saturation ratios

The dynamics of water uptake affects the composition profile within the particle, hence the equilibrium vapor pressure of water over its evolution. For instance, kinetic hindrance of water molecules moving between organic shell and aqueous core will lead to higher water content in the outer shell than one in which such hindrances do not exist. Concentration profiles of water within
the shell affect the water mole fraction and also droplet surface tension. Based on the findings from previous sections, three distinct scenarios through which LLPS may affect equilibrium saturation ratios of water are considered (Table 1). Given a fixed mass of dry substance, we use Köhler theory to calculate differences in equilibrium saturation ratios based on these assumptions how the driving force for water vapor condensation may be affected by chemical distribution and transport dynamics within an individual particle. Scenario (i) is valid at 300 K until the overall water concentration in the shell becomes too large for the

**Table 1.** Scenarios for Köhler theory calculations

| Scenario | Phases | Core/shell distr. of water | Shell composition | Surface tension |
|---|---|---|---|---|
| (i) | LLPS | kinetically hindered | nearly pure organic layer; nonuniform water profile | pure CPA[*] |
| (ii) | LLPS | kinetically hindered | mixed; uniform composition | saturated CPA solution[**] |
| (iii) | LLPS | kinetic hindrance ignored | mixed; uniform composition | saturated CPA solution[**] |
| (iv) | single | mixed; uniform composition | | pure water[†] |

[*] $\gamma = 30 \ \mathrm{mN \, m^{-1}}$ (Hyvärinen et al., 2006)

[**] $\gamma = 60 \ \mathrm{mN \, m^{-1}}$ (Hyvärinen et al., 2006)

[†] $\gamma = 72 \ \mathrm{mN \, m^{-1}}$ (Vargaftik et al., 1983)

gradient to prevent the formation of a well mixed aqueous layer at the surface. Scenario (ii) is a model of the 200 K behavior as well as that of 300 K from the point where the concentration gradient cannot prevent the presence of a non-negligible amount of water in the surface layer. Scenario (iii) is a hypothetical case that provides a lower bound on the water content in the shell. Scenario (iv) provides a base case where the distribution of species are considered to be homogeneous throughout the particle.

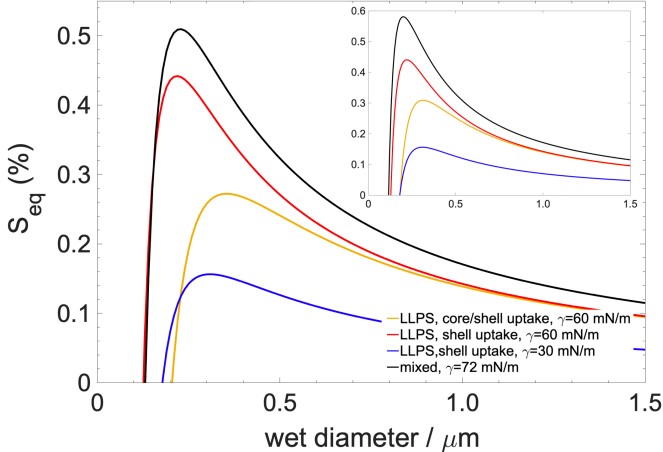

**Figure 7.** Köhler curves for 50 nm dry diameter particle containing 90% organic with 10% salt ($(NH_4)_2SO_4$). The inset shows Köhler curves for a dry particle which contains organics only. Black curve: well mixed case; blue curve: scenario 1 (hindered core uptake with concentration gradient); red curve: scenario 2 (hindered core uptake without concentration gradient); yellow curve: scenario 3 (unhindered core uptake without concentration gradient)

Köhler curves are calculated for particles having a 50 nm dry diameter consisting of 90% organic compounds and 10%
($(NH_4)_2SO_4$) and 100% organic with no salt (Figure 7). LLPS always tends to lower the critical supersaturation ($S_c$) and increase the critical diameter ($d_c$) compared to what is found for the well mixed case (iv). The lowering of the $S_c \sim 0.05$

% is the least pronounced if LLPS is preserved but only shell growth occurs, which is the scenario (iii) which describes the effect of hindered core uptake without concentration gradient. A moderate lowering of $S_c \sim 0.25\%$ is found for the scenario in which LLPS maintains moderate surface tension (the surface is a saturated mixture of h-CPA and water) with uniform core and shell growth, that non-hindered core uptake (ii). Finally, $S_c$ is the lowest if the concentration gradient is taken into account (by assuming a pure organic surface) (i), in this case the critical supersaturation is reduced by $\sim 0.35\%$. Trends are largely similar but more pronounced if the aqueous phase is assumed to contain only organics. In summary, LLPS — as a result of lower surface tensions characteristic of a saturated mixture — reduces critical supersaturations, while kinetic hindrance of the core uptake acts in the opposite direction since it increases the mole fraction of water in the organic shell. However, the concentration gradient and the resulting low surface tension characteristic of a pure or nearly pure organic surface imparts a strong effect, and invoke the strongest reduction in $S_c$ compared to the well mixed case among all the investigated scenarios.

## 4    Conclusions

Steered MD simulations of water uptake by a model LLPS aerosol particle consisting of a hydroxy-*cis*-pinonic acid surface layer and a pure aqueous core at two temperatures corresponding to the boundary layer (300 K) and to the top of the troposphere (200 K) were performed to investigate the mechanism of water uptake by LLPS aerosol via detailed analysis of the free energy profiles of the corresponding transfer process. In particular, the following questions were addressed: i) How does the uptake mechanism depend on temperature? ii) To what extent and under what conditions can water uptake by particles containing internal interfaces be described with a single uptake coefficient? iii) What role does the internal interface play in the water uptake mechanism? iv) How can the relationship between non-ideal mixing in LLPS particles and their increased CCN activity be explained on a molecular level?

These questions are answered using a novel combination of free energy profiles of interfacial transfer estimated from steered MD simulations and intrinsic surface analysis, which removes artificial smoothing and systematic errors caused by thermal fluctuations of liquid surfaces from estimates in interfacial properties and density profiles. Free energy profiles together with their entropic and energetic contributions were used to determine the water uptake mechanism, map the effect of the presence of an internal surface on the shape of the free energy, energy and entropy profiles and identify main thermodynamic driving forces behind the observed mechanism. Using steered MD in this context presents an advancement over previous approaches. The choice of the method can be rationalised by considering that each finite velocity pulling simulation used for estimating the free energy based on Jarzinsky's equality closely mimics one realistic interfacial mass transfer event. The ensemble averaging ensures that a variety of potential pathways is included in the free energy estimate. Implications for realistic atmospheric processes were presented in the form of model calculations that link the molecular scale mechanism to the increased CCN activity of LLPS aerosol quantitatively for our model system and qualitatively for generic LLPS particles.

Our findings can be summarized as follows: i) The mechanism of water uptake (the shape of the free energy profile) is strongly temperature dependent. All minima and maxima can be attributed to entropic contributions, the minimum at the organic/water interface is due to a local maximum of the conformational entropy, while the subsequent maximum is a con-

sequence of increased ordering of the water molecules in the first molecular layer of the aqueous phase. Due to the explicit temperature dependence of the weight of the entropy term in the free energy profile, features disappear at low temperature, however structural properties shaping entropy profiles are generally present in liquid/liquid and liquid/solid interfacial systems arbitrary composition. ii) Mass accommodation (vapor-to-organic transfer) coefficients were found to be near unity at both temperatures, which is in accordance with globally representative values. The core uptake (organic-to-water transfer) coefficient

is reduced at room temperature ($k_{o/w}$=0.03), while at low temperature $k_{o/w}$=1. This suggests that a single uptake coefficient is sufficient to describe the water uptake mechanism at 200 K, while core uptake might have to be taken into account for the higher temperature. iii) Model estimates of shell and core uptake timescales revealed that depending on the particle phase diffusion coefficient shell uptake is at least four orders of magnitude faster that of core transfer. The slow diffusion of water in the organic shell can cause water to accumulate in the shell. This difference is further increased by 1-2 orders of magnitude

if core uptake coefficient as a result of interfacial ordering of water molecules, leading to even more retention of water in the organic shell. iv) Converting free energy profiles into diffusion corrected concentration profiles allowed us to determine how molecular scale non-idealities in the solution structure can lead to enhanced surface activity. The molecular-scale explanation of the effect of non-ideal mixing on CCN activity lies in a non-uniform distribution of water molecules within the organic shell observed for liquid particles at 300 K. The concentration distribution has a maximum near the organic/water interface,

indicating that the condensing water molecules tend to accumulate near the aqueous phase and leaves the surface depleted in water. In other words, the observed concentration gradient maintains low surface tensions (nearly pure organic surface) and the LLPS state even when RH approaches 100%. Köhler calculations reveal consequent reduced surface tensions are able to compensate the unfavorable effect on hindered core uptake on critical supersaturations.

     In summary, our results point out that a single uptake coefficient is sufficient to describe water uptake in LLPS aerosol at

low temperature, while at room temperature the models based on the complete uptake mechanism might be preferred. The effect of non-ideal mixing — usually accounted for in the form of Flory-Huggins parameter in continuum model calculations — are attributed to non-uniform distributions of the condensing water which maintain surface tension at low values even at high RH. The generalizability of thermodynamic driving forces suggests that the development of detailed models of aerosol growth kinetics incorporating these findings is possible, when combined with more rigorous and quantitative studies.

The strong temperature dependence of the water uptake mechanism, core uptake coefficient, as well as the presence non-uniform distribution of water within the organic shell at room temperature suggest that a detailed description of water uptake including these effects in a temperature dependent manner is necessary to improve aerosol growth kinetic models. The driving forces responsible for the typical features of the free energy profiles are generally valid for a wide range of liquid/vapor,liquid/liquid and liquid/solid surfaces. They presumably depend only weakly on the chemical nature of the organic

compounds, which suggest that developing such a parametrization is feasible.

## 5   Data availability

PLUMED input files and starting configurations are available upon request. A movie showing a sample steered MD trajectory at 300 K is available at https://zenodo.org/record/4902870#.YLsZwS0Rq_U.

*Video supplement.*

## Appendix A:  Structural analysis

### A1   Steered molecular dynamics

In Figure A1 we demonstrate through the example of 3 randomly selected realizations from both temperatures, that the steered MD trajectories can result in an arbitrary impact angles. In the equilibrium snapshots the position of the pulled molecule is shown in the initial configuration and at the point of contact formation with the particle. The organic phase in the snapshots

shows the configuration at the moment of impact but its center of mass has been aligned with that of the initial configurations to ensure that the impact angles are correctly represented.

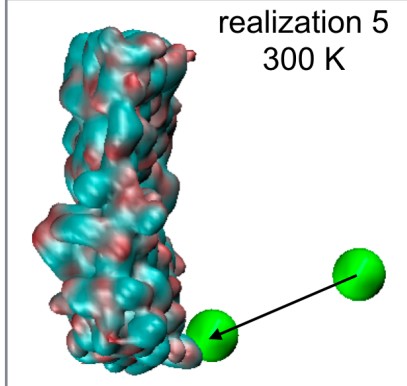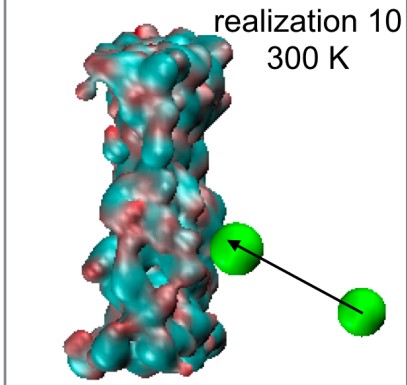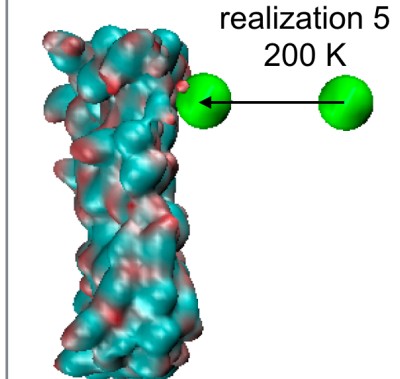

**Figure A1.** Snapshots showing the impact angle in randomly selected trajectories. The pulled molecule is shown as a green bead while the organic phase is represented by it solvent accessible surface. The aqueous phase was removed from the snapshot for clarity.

### A2   Intrinsic surface analysis and intrinsic density profiles

We label molecules of the organic/water interface in a time resolved manner using the ITIM method (Sega et al., 2018). ITIM selects interfacial molecules by solely geometric criteria, thus it is considerably faster than alternative methods with essentially

no loss in accuracy (Jorge et al., 2010). The ITIM method uses a probe sphere with a radius determined from the position of the first peak of the corresponding radial distribution functions, in our case a value of $r = 0.2$ nm is used. The probe sphere is moved along a grid of testlines (a $200 \times 200$ grid is used in our analysis) perpendicular to the macroscopic plane of the interface. Once the probe sphere touches a an atom, the molecule to which it belongs to is labeled as interfacial. Contact is determined based a Pythagorean criterion. The list of surface molecules allows for selective estimation of various properties of

the surface and the bulk and provides a means to reconstruct the intrinsic density profiles:

$$\rho(z) = \frac{1}{A} \left\langle \sum_i \delta_i(z - z_i) + \xi(x_i, y_i) \right\rangle, \tag{A1}$$

where $A$ is the macroscopic surface area of the interface, $x_i, y_i$ and $z_i$ are the Cartesian coordinates of the atoms constituting the system and $z$ is the position with respect to the local interface. $\xi(x_i, y_i) \sim kT/q^2$ is the capillary wave mode spectrum, with $q$ being the wave vector. In simple terms intrinsic density profiles are anchored to the first molecular layer of one of the condensed

phase (the opposite phase in our case), instead of being calculated along an external grid, thus they are able to resolve the near-surface fine structure of the density profiles which are otherwise washed away by capillary wave fluctuations. Intrinsic number density profile are used to interpret free energy profiles in a qualitative manner. The ITIM algorithm in particular allows for the separation of the surface molecules from those belonging to the bulk, and thus repeating the algorithm on the remaining bulk phase molecules can yield consecutive subsurface layers. Separate layers are used to estimate layer-by-layer density profiles as

well as orientational distributions within the first two molecular layers of the aqueous phase.

Using intrinsic surface analysis always reveals the layered structure of the near-surface region. Water at room temperature tends to have a very slight layering, with hardly any structure noticeable beyond the second peak(Jorge et al., 2010). Our results show an even less pronounced structure, due to the mixed nature of the interface, which is explained in more details in the next section. In order to shed light on the enhanced structure that appears only in the intrinsic density profiles intrinsic and global

mass density profiles of water are overlapped (Figure A2).

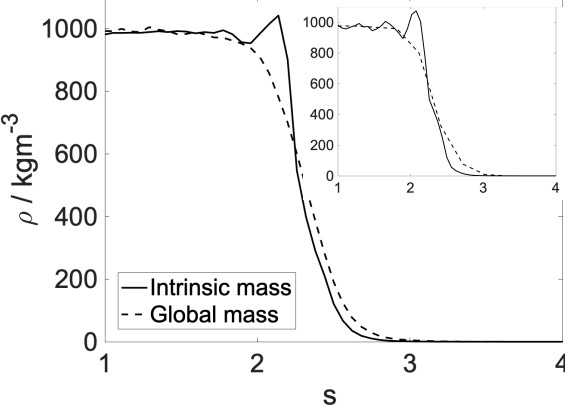

**Figure A2.** Intrinsic (solid line) and global (dashed line) density profiles of the aqueous phase. Main panel: 300 K, inset: 200 K

## A2.1 Analysis of the interface structure

In order to clarify the origin of the density drop, which is a non trivial phenomenon, at least from a macroscopic perspective, we present analysis that highlights the atomistic structure of the organic/water interface. As mentioned before, liquid surfaces are corrugated by capillary waves. These corrugations cause the density drop between the two bulk phases(Jedlovszky et al., 2002; Fernandes et al., 1999; Benjamin, 1997). Depending on the nature of the organic phase (its solubility in water, and the intermolecular interactions that it forms with water molecules), the organic/water interface can either be sharp fully phase separated (eg.: isooctane or heptanone)(Fernandes et al., 1999), or mixed. In the former case the density drop is clearly related to the differing fluctuations of the two sharp surfaces, causing the formation of a thin gap between the two phases. For mixed or interpenetrated interfaces, this effect is somewhat reduced by mixing of the two phases, however voids are still observable as a result of capillary wave fluctuations. Examples of such voids are shown in A3 a). In order to determine whether a liquid/liquid interface is sharp or interpenetrated one has to examine the sign of the local interface width $W_{ij}(z)$ in elements of a fine perpendical grid that divides the interface into small elementary units, using the following formula:

$$W_{ij}(z) = max(L_{ij}(z)) - min(R_{ij}(z)), \tag{A2}$$

where $L_{ij}(z)$ and $R_{ij}(z)$ are the coordinates of the atoms of the left and right phase in the $ij^{th}$ gridcell respectively. A negative sign of the mean width trivially means sharp phase separated surfaces while a positive sign means that the interface is mixed. In this work we use a rectangular grid along the the surface normal that divides the XY plane into elementary units of the area of 0.04 nm$^2$, the left phase is the aqueous phase and right phase is the h-CPA. Figure A3 b) show the histogram of $W_{ij}(z)$ obtained from an equilibrium MD trajectory (3 ns) of the interfacial system, representative of the equilibrium structure of the surface in the absence of a transferred molecule. It is clear that the interface in our system is of the mixed type ($W_{ij}(z)$ assumes almost exclusively positive values). This is consistent with the intrinsic density profile not being exactly zero in the interfacial region. For fully phase separated system, the intrinsic density profiles decrease steeply to zero in between the two constituting phase(Jorge et al., 2010).

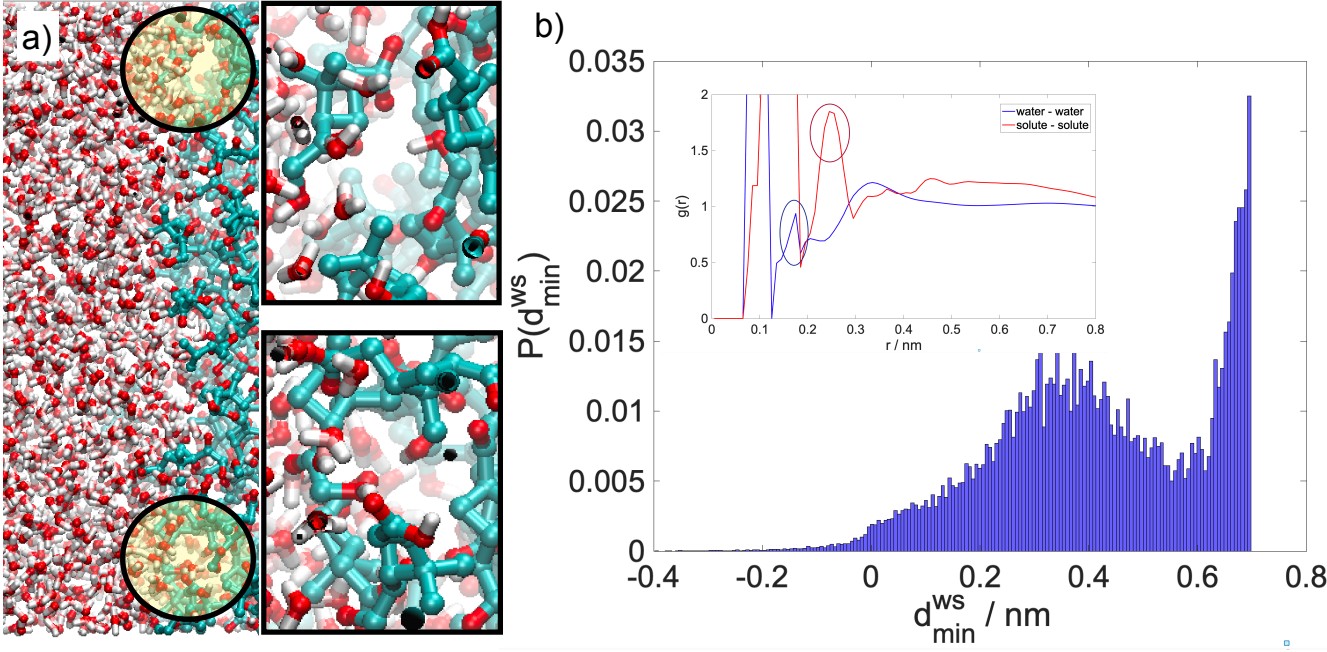

**Figure A3.** a) Equilibrium snapshot of the water/organic interface with examples of voids enlarged. The probability distribution of the minimum local distance between the maximum of the aqueous phase and the minimum of the organic phase at the interface in the absence of the pulled molecule. For comparison in the inset we show bulk radial distribution functions whose first intermolecular peaks (circled) can give an estimate of the mean distances in the bulk phases.

It's worth noting that the distribution is bimodal, indicating the existence of smaller and larger voids between the two phases. When compared to the mean distance between nearest neighbor molecules in the bulk phase, estimated as the position of the first intermolecular peak of the corresponding radial distribution function (Inset of Figure A3 b)), these mean minimal distances are non-neglibibly larger. Thus voids in the interface, defined here as the region bounded by the position of the peaks of the intrinsic density profiles, tend to be larger than voids found in any of the bulk phases, which leads to a decrease of the total density. Additionally, as shown in Appendix B, these voids do not reduce the number of hydrogen bonds of the transferring molecule, and thus provide sterically and energetically favorable environment for the pulled water molecule.

## A2.2 Interface layering

The density profile of the first 2 molecular layers detected by the ITIM algorithm are overlapped here with the global intrinsic density profiles.

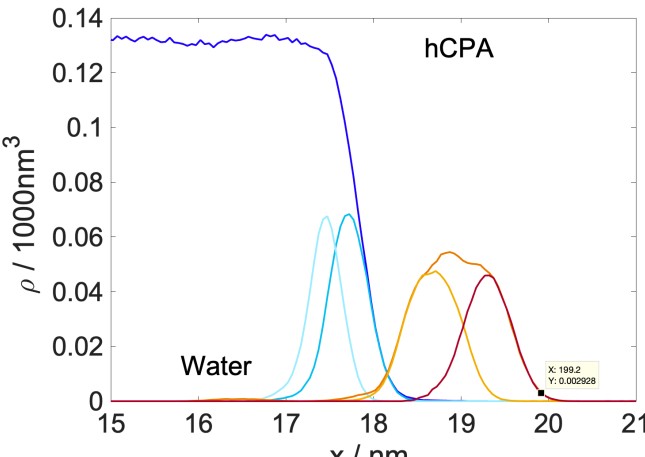

**Figure A4.** Global number density profiles overlapped with the density profiles of the first two molecular layers detected by the ITIM algorithm. Density profiles are not centered at the center of mass of the aqueous phase.

The width of the first layer of the aqueous phase is 0.48 nm , that of the organic phase 0.7 nm (from the aqueous side) and 0.6 (from the vapor side), measured at half of the height of curves. The large overlap between the first and second molecular layer in the aqueous phase explains the very weak structuring of the subsurface layers. The organic phase is composed of two overlapping interfacial layers, and its width is not sufficient to accommodate any additional layers not in direct contact with any interfaces.

### Appendix B: Thermodynamic Analysis

The free energy profiles were decomposed into energetic and various entropic contributions in order to understand the effects responsible for the features observed in the in the free energy profiles

### B1    Internal energy and hydrogen bonding

The internal energy profile of the transfer process is estimated as the sum of the interaction energy between the pulled water molecule and the organic and water molecules weighted by the local mole fraction of the above two.

$$\Delta E_i(s) = x_s^l(s) * E_i(p,s) + x_w^l(s) * E_i(p,w), \tag{B1}$$

with $E_i(p,s)$ and $E_i(p,w)$ being the interaction energy between the pulled molecule and the solutes/water, calculated as the sum of short-term Coulombic and Lennard-Jones interactions (example values are listed in Table B1). $x_s^l(s)$ and $x_w^l(s)$ are local mole fraction profiles of the water and the organics. Local mole fraction profiles are calculated from the number of water and organic molecules found within 1 nm of the pulled molecule. The cutoff distance of 1 nm corresponds to the cutoff used in the simulations for short range interactions. This calculation is only plausible because solute/water interactions in the OPLS

 and TIP4P force fields were parametrized partially on quantum chemical calculations, thus individual interaction energies are physically meaningful. Equation B1 is evaluated for all realizations and averaged to yield the final profiles. Internal energy profiles of selected realizations are shown in Figure B1.

**Table B1.** Coulombic and Lennard-Jones contributions of the interaction energies in selected realisations

| | R1 | | R2 | | R3 | |
|---|---|---|---|---|---|---|
| Type | $200K$ | $300K$ | $200K$ | $300K$ | $200K$ | $300K$ |
| LJ(p,s) | 0.2 (0.1) | -0.3 (0.07) | -0.43 (0.2) | -0.2 (0.02) | -0.09 (0.02) | -0.09 (0.06) |
| Coulomb(p,s) | -5.7 (1.3) | -3.4 (1.1) | -5.4 (2.1) | -3.7 (1.2) | -6.8 (3) | -3.8 (1.3) |
| LJ(p,w) | 14.4 (1.9) | 11.3 (1.5) | 15.1 (2.2) | 11.1 (1.4) | 14.2 (2.1) | 10.9 (1.4) |
| Coulomb(p,w) | -74.5 (10.5) | -66.5 (8.5) | -75.7 (10.4) | -66.4 (8.2) | -71.8 (10.2) | -65.6 (1.0) |

The formation of hydrogen bonds is a major energetic driving force of the water uptake process. Figure B2 shows the number of total hydrogen bonds along the direction of the reaction coordinate.

## B2 Entropy

### B2.1 Interfacial entropy

Interfacial entropy accompanying any molecular transfer across phase boundaries can be calculated using the following formula from statistical thermodynamics (Ward, 2002):

$$\frac{\Delta S_{IF}}{k} = 4\left(1 - \frac{T_L}{T_V}\right) + \left(\frac{1}{T_L} - \frac{1}{T_V}\right)\sum_{l=1}^{3}\left(\frac{\hbar\omega_l}{2k} + \frac{\hbar\omega_l/k}{exp(\hbar\omega_l/kT_V) - 1}\right) +$$

$$+ \frac{v_L^{sat}}{kT_L}[P_V - P_{sat}(T_L)] + ln\left[\left(\frac{T_L}{T_V}\right)^4\left[\frac{P_{sat}(T_L)}{P_V}\right] + ln\left[\frac{q_{vib}(T_V)}{q_{vib}(T_L)}\right]\right], \quad \text{(B2)}$$

where $T_V$ and $T_L$ are the temperatures of the vapor and the liquid phase, $P_{sat}$ and $P_V$ are the saturated and the actual vapor pressure, $v_L$ is the specific volume of the liquid phase $\omega_i$ are the vibrational frequencies and $q_{vib}$ is the vibrational partition function. Figure B3 shows the modeled $T\Delta S_{IF}$ profiles at the two simulated temperatures, the values corresponding to the vapor pressures in the simulation box are highlighted with asterisks. The vapor pressure in the simulation box is estimated assuming the presence of the pulled molecule only in the vapor phase, using the universal gas law.

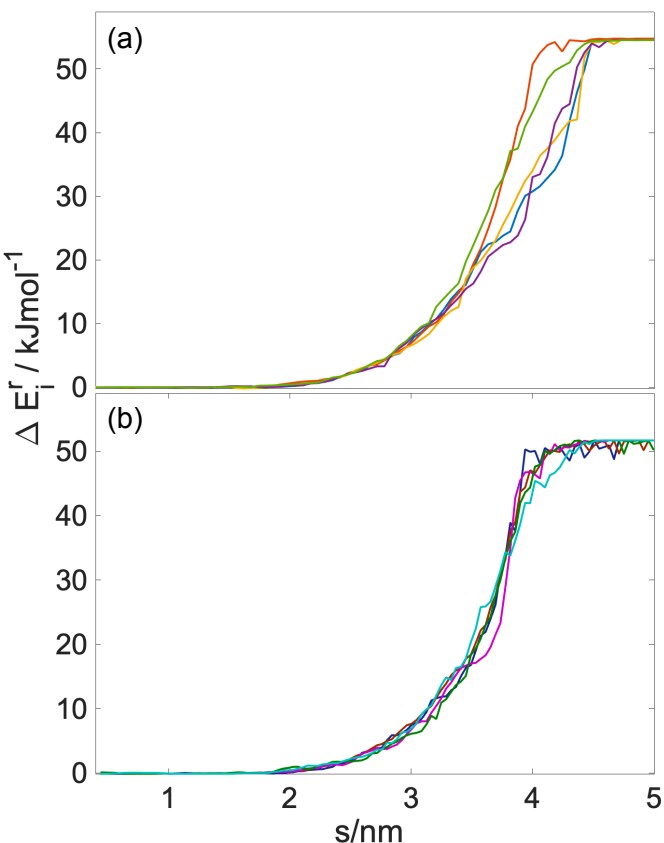

**Figure B1.** Internal energy profiles of selected realisations, a) 200 K, b) 300 K.

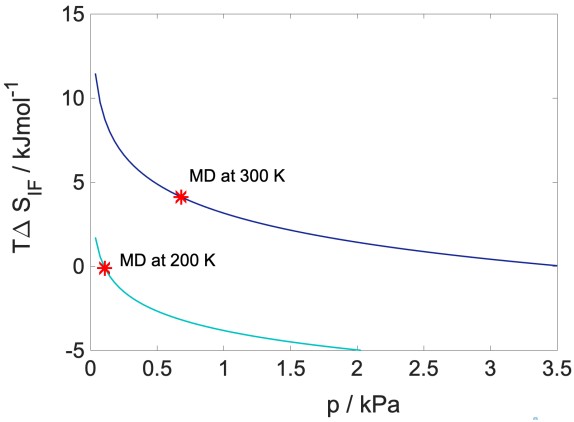

**Figure B3.** Modeled interfacial entropies at the simulated temperatures

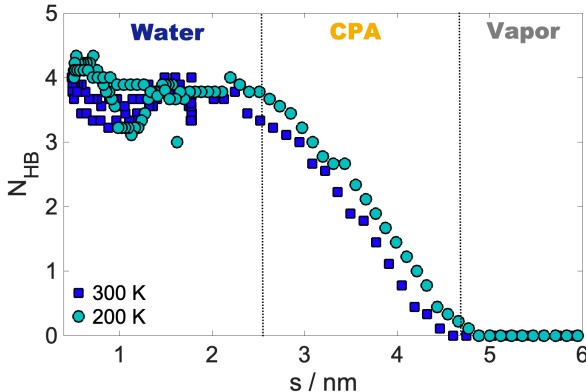

**Figure B2.** The number of hydrogen bonds formed by the pulled molecules along the reaction coordinate

### B2.2  Conformational entropy

The conformational entropy profile is estimated as:

$$S_{conf} = -k_B \sum_i x_i(s) ln(x_i(s)),$$
(B3)

where $x_i(s)$ denotes mole fraction profiles of the components of the systems, with $s$ being the reaction coordinate used in the steered MD simulations.

### B2.3  Configurational entropy

We calculate configurational entropy according to Schlitter's formula (Baron et al., 2006), thus based on the covariance matrix ((D)) of the atomic coordinates between two distinct groups of atoms, one being the pulled molecule and the other is either the
705 ensemble of the solutes or waters constituting the bulk phase.

$$S_{config} = \frac{k_B}{2} \ln \det \left( 1 + \frac{k_B T e^2}{\hbar} \mathbf{D} \right),$$
(B4)

where $\hbar$ is the Planck's constant divided by $2\pi$ Two different contributions of the configurational entropy are considered i) between the pulled molecule and the solutes and between the pulled molecule and the solvents. In similar manner as for the internal energy, the weighted sum of these two yields the configurational entropy profile along the reaction coordinate ($s$), with
710 the weights being the local mole fractions of the water and the solute, whose calculation is described in the previous section.

### B2.4  Orientational entropy

We propose an equation which can serve as a qualitative descriptor of the entropy related to the orientation of the molecules based on equations for translational (Bhandary et al., 2016) and translational-orientation entropy terms. (Piaggi and Parrinello,

2018). With a simple exchange of the radial distribution function in (Bhandary et al., 2016) with the angular distribution function ($g(\theta)$) of the angle ($\theta$) between the dipole vector of the water molecules and the surface normal axis, we obtain an expression for orientational entropy:

$$S_{or} = -2\pi k_B \int \left[ g(\theta) lng(\theta) - g(\theta) + 1 \right] sin\theta d\theta \qquad \text{(B5)}$$

This expression is evaluated for angular distribution functions calculated in the first two molecular layers and the bulk of the aqueous phase, to highlight the effect of increased molecular order on the free energy profiles. The separation of interfacial water molecules and those constituting the second layer is performed by two consecutive repetitions of the ITIM algorithm, using the output bulk phase of the first one as input for the second one.

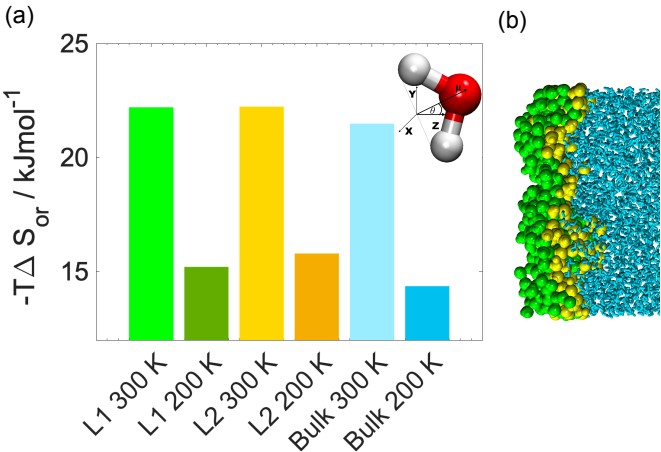

**Figure B4.** a) Orientational entropies in the first and second molecular layers and the bulk phase for both simulated temperatures. Inset: the definition of the $\theta$ angle. b) Snapshot from a 300 K realisation showing the first two molecular layers of the aqueous phase in green (L1) and yellow (L2).

At both temperatures the bulk phase orientational entropy is higher ($-T\Delta S$ is lower) than in the first two layers due to stronger ordering in the first two interfacial layers (Figure B4). We note that equation B5 cannot completely describe the entropy loss due to preferential ordering of the water molecules at the interface since due to its point group symmetry ($C_{2v}$), the orientation of water molecules with respect to an external vector or plane cannot be described with a single angle, instead the joint distribution of two angles is necessary. The development of an adapted expression of the orientational entropy of such cases is however out of the scope of this study. The one dimensional representation is incomplete and thus gives only a qualitative insight but the temperature dependence of $T\Delta S_{or}$ within the layers is clear, the extension of the orientational entropy formula to multiple dimensions is part of ongoing work. To complete the description of orientational differences between the surface and the bulk of the aqueous phase, we calculate joint distributions of angles $cos\theta$' and $\phi$, which are chosen to fully describe the orientation of water molecules with respect to the normal vector of the macroscopic surface(Bartók-Pártay et al., 2008), in the first two molecular layers and the bulk. $\theta$' and $\phi$ are defined in a Cartesian frame centered on the water

molecules, the **z** axis points from the water oxygen towards the midpoint of the segment connecting the hydrogen atoms, the **y** axis is parallel to that segment, and the **x** axis is perpendicular to both **z** and **y**. $\theta$' is the angle between the macroscopic surface normal vector (**Z**) and the molecule centered **z** axis, and $\phi$ is the angle between the **x** axis and the projection of the surface normal vector to the **x,y** plane.

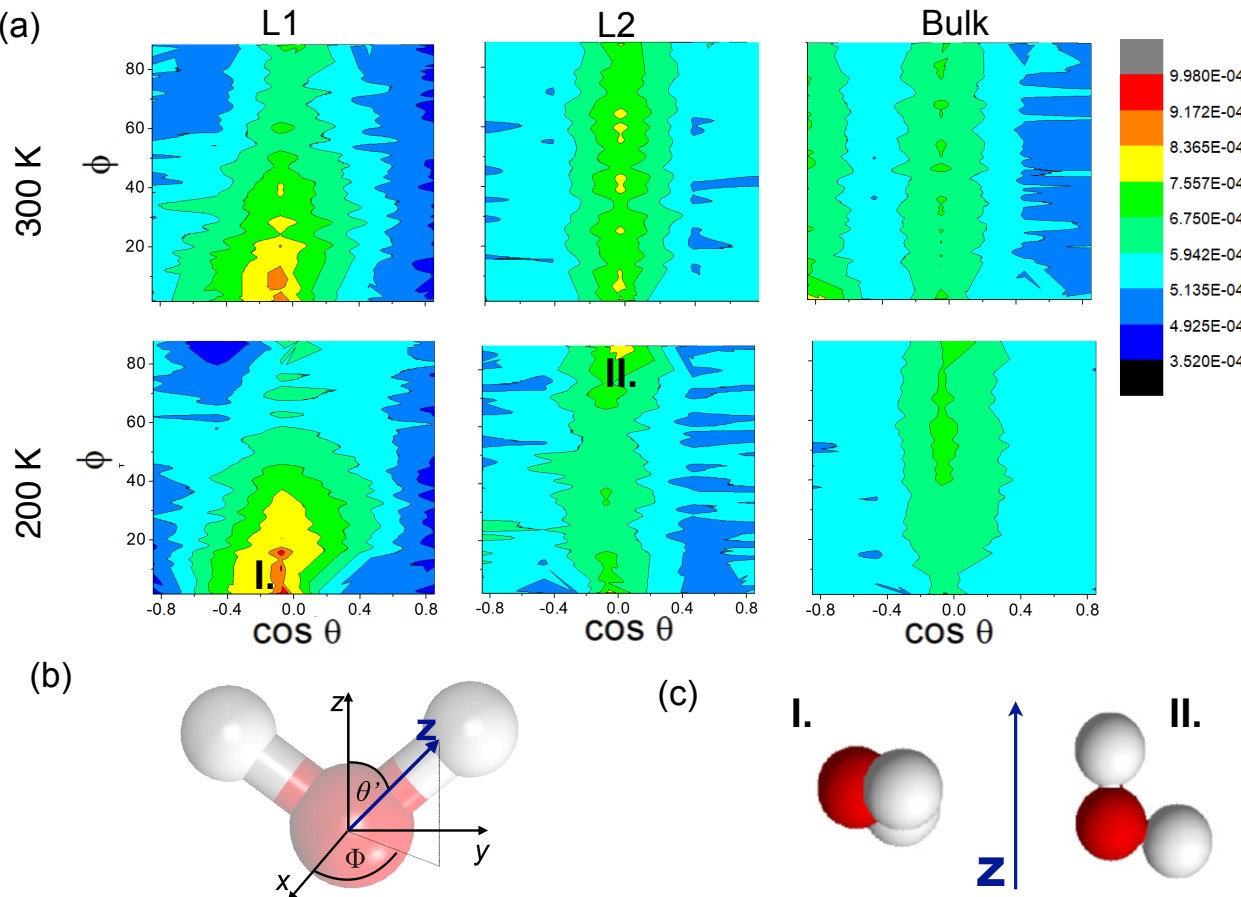

**Figure B5.** (a) Orientational maps of the water molecules in the first (L1), second (L2) molecular layer and the bulk phase. (b) The definition of $\theta$' and $\phi$ in the Cartesian frame centered on a water. (c) Examples of the two main orientations.

Figure B5 shows orientational maps in the first two molecular layers and the bulk of the aqueous phase. In the first layer water molecules show a very strong preference to be aligned with dipole vectors parallel to the surface or slightly tilted towards the bulk aqueous phase, orientation I. in Figure B5 a) and c). In another distinguished orientation, which appears in the first and second layer at 300 K and the second layer at 200 K, one O-H bond points in the direction of the surface normal vector and the other slightly inwards to the bulk aqueous phase. Orientational preferences diminish progressively when moving towards the bulk phase, indicating the decrease of orientational order and thus a increase in orientational entropy. It is

a remarkable difference between our system and aqueous interfaces of hydrophobic organic compounds (dichloromethane and dichloroethane) studied previously (Hantal et al., 2010), where preferred orientations were only found in the first molecular

layer of water in direct contact with the organic phase, that the second layer is more ordered than the bulk phase, it is due to the fact that h-CPA mixes more readily with water then hydrophobic organics, thus h-CPA molecules can penetrate into the second and third molecular layer as well into the bulk phase, and contact with the dissolved organic molecules promotes orientations that are similar to those found at the interface. We also note that the preferred orientations found of interfacial waters are universal across a large spectra of organic/water interfaces.

## Appendix C: Statistical Analysis

### C1    Statistical accuracy of the results

The statistical accuracy of the free energy profiles is assessed using an approach inspired by bootstrapping, which has already been used to analyse the statistical accuracy as a function of sample size for steered MD simulations(Potterton et al., 2019). Free energy profiles are calculated on subsets of the whole sample size containing a varying number of realizations, created by

removing $n$=1-22 randomly selected trajectories from the total number of samples. 10 different random samples of each size are generated and then used to estimate mean free energy profiles, their standard deviation and 95% confidence intervals at each sample size.

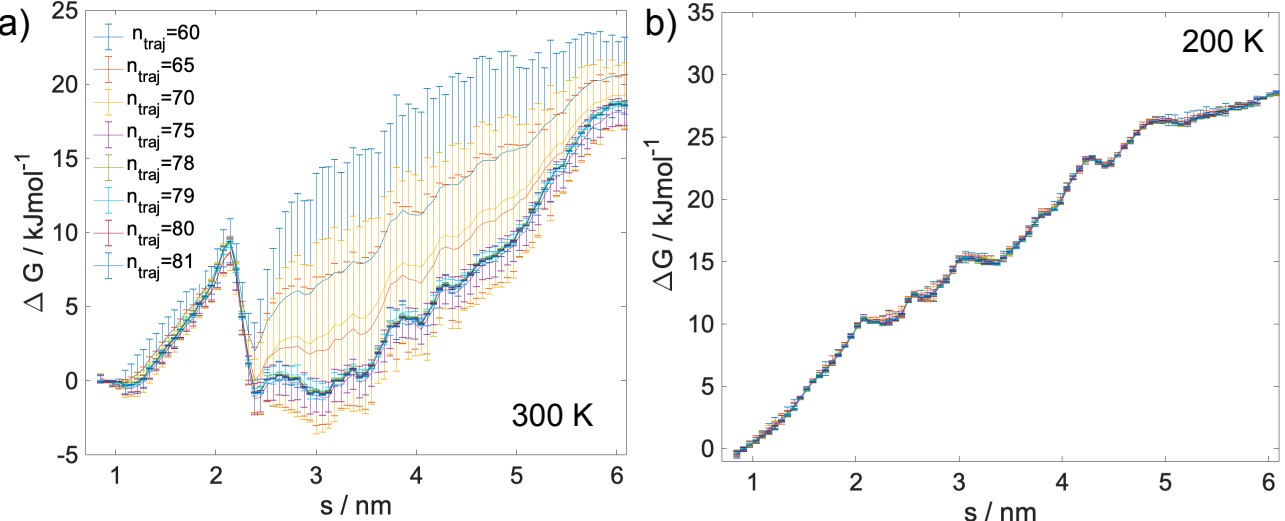

**Figure C1.** Mean free energy profiles at varying sample size at a) 300 K and b) 200 K. Error bars represent one standard deviation.

Figure C1 a) shows the free energy profiles at 300 and 200 K estimated from different numbers of samples. At 300 K the free energy profiles become nearly invariant with the number samples for samples sizes superior to 75, above which also the standard deviation of the free energy in each point of the profile becomes considerably smaller than for smaller samples. Smaller number of samples suffice to produce converged free energy profiles at 200 K.

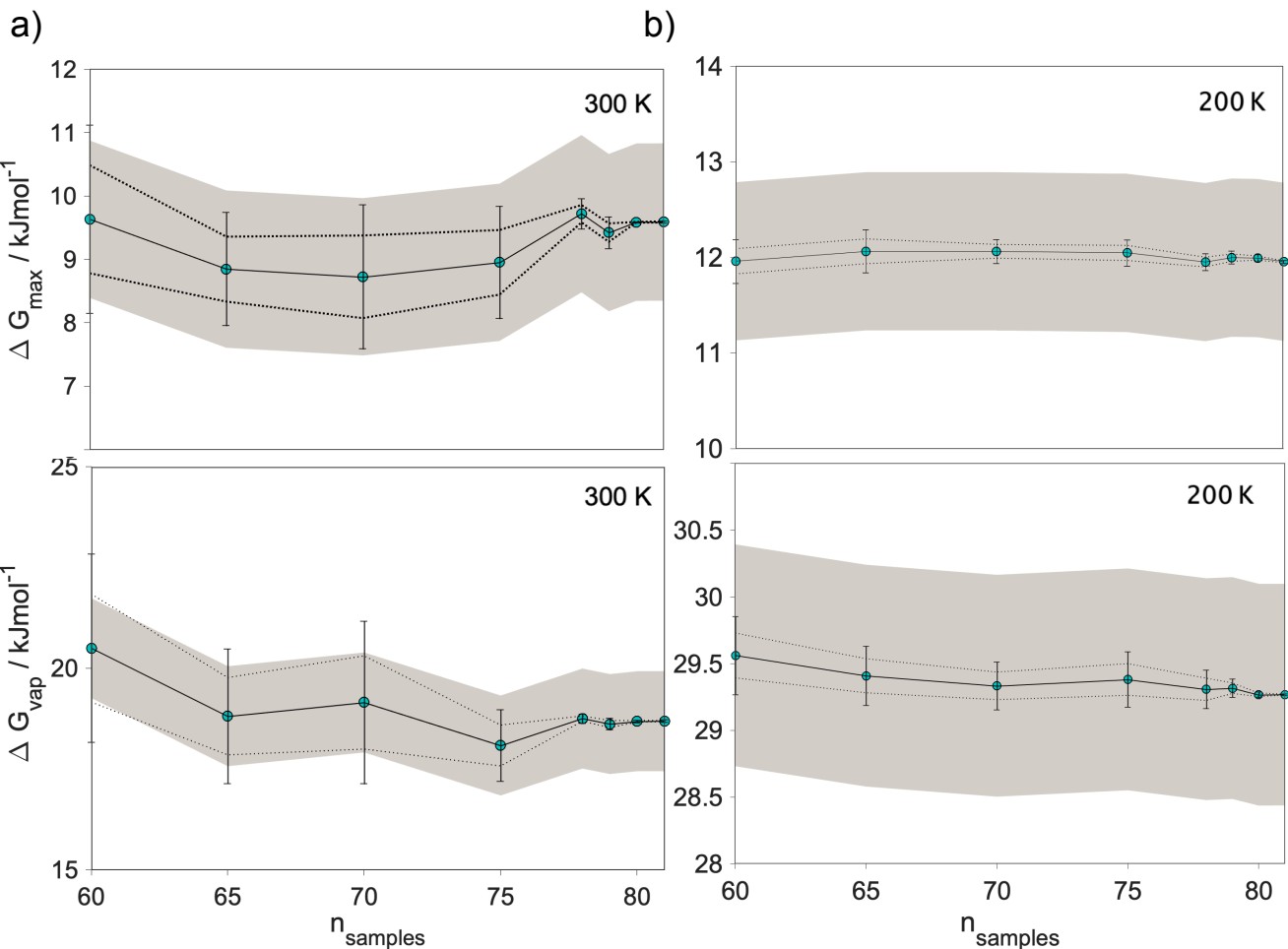

**Figure C2.** a) Top panel: value of the free energy in the vapor phase at 300 K. Bottom panel: the maximum of the free energy in the bulk liquid phase at 300 K b) Top panel: value of the free energy in the vapor phase at 200 K. Bottom panel: the maximum of the free energy in the bulk liquid phase at 200 K. The error bars represent one standard deviation, dotted lines are the 95% confidence intervals and the grey shaded area indicates the region that falls within $\pm 1/2 k_B T$ from the mean values.

Figure C2 shows the value of the free energy in the vapor phase and at the free energy maximum at 200 K and 300 K as a function of sample size, with their 95% confidence intervals and their standard deviation to assess statistical accuracy by comparing them against $k_B T$ on the given temperature. In classical molecular simulations the energy of the thermal motion

$(3/2k_BT)$ is often considered as the limit of statistical significance. Here we compare to a stricter criterion, i.e. we compare the changes of the mean values with the sample size, their standard deviation and their 95% confidence interval to the value of the Boltzmann factor and accept the free energy as statistically accurate if the changes in the mean value upon increasing the sample size are smaller than $k_BT$ and both the standard deviation and the confidence interval fall within a range of $\pm 1/2k_BT$ from the mean value of the free energy $\langle \Delta G \rangle$. According to this assessment at 300 K the number of samples necessary to

reproduce statistically significant vapor phase free energy values is 75, for the maxima 65 samples are sufficient. At 200 K these criteria are fulfilled already at the smallest sample size containing 60 realisations. From this analysis we conclude that our free energy profiles estimated from 82 realisations can be considered converged and statistically significant within a range of uncertainty that can be attributed to thermal motion at the temperature of interest.

## C2    Thermodynamic consistency of the data

### C2.1    Vapor phase internal energy and enthalpy of evaporation

The heat of vaporization, calculated as the difference of the internal energy of the vapor and the aqueous phase is $\sim 55\,\mathrm{kJ\,mol^{-1}}$ at 300 K and $\sim 60\,\mathrm{kJ\,mol^{-1}}$ at 200 K, the latter extrapolated from the lowest available temperatures (240-250 K) assuming a linear fit (from the Dortmund Data Bank, 2021, www.ddbst.com). The latent heats of vaporization of pure water, $43.9\,\mathrm{kJ\,mol^{-1}}$ at 300 K and $58\,\mathrm{kJ\,mol^{-1}}$ at 200 K, are within 25 and 4% from the simulated values. This is not a direct comparison, since

the presence of the organic layer potentially impacts the latent heat of vaporization to an unknown extent, however the trend is well reproduced and the percentile differences indicate a semi-quantitative agreement with the experimental values.

### C2.2    Comparison with experimentally derived free energies of transfer

The difference between the vapor and liquid phase end of the simulated free energy profiles corresponds to the free energy change related to the transfer of a single water molecule from the liquid phase to the vapor phase ($\Delta_v^l G$, the free energy change

that accompanies the evaporation of a single molecule). The free energy of transfer between two phases of different density can be estimated from the density of the transferred molecule in the liquid ($\rho_l^t$) and the vapor phase ($\rho_v^t$))(Ben-Naim, 1978; Wick et al., 2003; Martin and Siepmann, 1998). For evaporation the eqaution takes the form of

$$\Delta_v^l G = -RT \ln \frac{\rho_v^t}{\rho_l^t}. \tag{C1}$$

In our model calculation $\rho_l^t$ is the density of the pure water at the temperature of interest. To the best of our knowledge, at 200

K measurements of experimental densities of neither supercooled water nor ice are available, therefore the values of the lowest measured temperatures are temperature are used in the calculation. $\rho_v^t$ is simply estimated from the universal gas law as $p/RT$, assuming that water vapor behaves as ideal gas under atmospherically relevant conditions. The vapor pressure of supercooled water and ice is estimated from parametric equations(Wexler et al., 1977; Mauersberger and Krankowsky, 2003; Murphy and Koop, 2005). Table C1 summarizes the data used in the calculations and the obtained free energy differences.

**Table C1.** Estimates of the free energy of transfer based on experimental data

| condensed phase | T / K | $p_v$ / Pa | $\rho_v^t$ / $m^{-3}$ | $\rho_l^t$ / $m^{-3}$ | $\Delta_v^l G$ / kJ mol$^{-1}$ |
|---|---|---|---|---|---|
| liquid water | 300 | 353 | $0.85 \times 10^{23}$ | $33.23 \times 10^{27}$ | 26.35 |
| supercooled water | 200 | $0.5^a$ | $18.04 \times 10^{19}$ | $32.60 \times 10^{27 b}$ | 31.56 |
| ice | 200 | $0.26^c$ | $9.38 \times 10^{19}$ | $30.76 \times 10^{27 d}$ | 32.54 |
| ice | 200 | $0.46^e$ | $16.59 \times 10^{19}$ | $30.76 \times 10^{27}$ | 31.60 |

[a] vapor pressure over supercooled water taken from the parametrization of Murphy and Koop(Murphy and Koop, 2005) [b] density of supercooled water at 239 K(Liu et al., 2008) [c] vapor pressure over ice taken from the parametrization of Wexler(Wexler et al., 1977) [d] density of ice at 223 K (Raznjevic, 1976) [e] vapor pressure over ice taken from the parametrization of Mauersberger(Mauersberger and Krankowsky, 2003)

From our simulations $\Delta_v^l G = 19.7$ kJ mol$^{-1}$ at 300 K and 29.4 kJ mol$^{-1}$ at 200 K. Our simulations systematically underestimate the free energy differences predicted from Equation C1 by 7 kJ mol$^{-1}$ and by 2 kJ mol$^{-1}$ at the two temperatures respectively. However, given that the statistical errors are much smaller than the difference between $\Delta_v^l G$ at the simulated temperatures, we can conclude that the temperature dependence is well reproduced.

Gibbs Ensemble Monte Carlo simulations yield 23 kJ mol$^{-1}$ at 300 K and 25 kJ mol$^{-1}$ at 273 K for pure TIP5P water(Wick et al., 2003). They conclude that the underestimation of the free energy of transfer at a given temperature and the overestimation of the temperature dependence are caused by the fact that the TIP5P water model, optimised to reproduce specific density at the density maximum and the liquid structure at room temperature, cannot reproduce the free energy of transfer in a quantitative manner(Wick et al., 2003).Umbrella-sampling simulations(Ergin and Takahama, 2016) yield a value of $\Delta_v^l G$=24 kJ mol$^{-1}$ for the SPC/E water model, which is known to underestimate the vapor pressure at room temperature(Vega et al., 2006). The TIP4P water model is among the best to reproduce hydrogen bonding in the condensed phase(Zielkiewicz, 2005), as well as the vapor pressure of water as a function of temperature(Vega et al., 2006). However, it still overestimates the experimental vapor pressure by a factor of ∼1.2, which may slightly contribute to the underestimation of the free energy differences. More likely the observed difference is due to the presence of the organic coating and the fact that the aqueous phase of our system at 300 K is not pure water but an aqueous solution of h-CPA, which can contribute to the difference between experimental and simulated values. Since we use a very simple transferable potential to represent the organic compound, we cannot expect our simulations to provide a quantitative estimate of the free energy of transfer.

In conclusion, we cannot reach quantitative agreement between our simulations and the free energy differences estimated from the vapor pressure and the liquid phase density. This partially due to limitations of the force field and partially to the fact that no direct comparison can be made between the aqueous phase in our simulation at room temperature and pure water. However, trends are well reproduced and a semi-quantitative agreement with experiments is achieved, which is sufficient to support our conclusions.

*Author contributions.* M.D designed and performed simulations and analysis and wrote the manuscript. S.T. and A.N. contributed to analysis, interpreted results and wrote the manuscript.

*Competing interests.* The authors declare no competing interests

*Disclaimer.* TEXT

*Acknowledgements.* We acknowledge funding from the Swiss National Science Foundation (Project numbers: CRSK-2_195329 and SNF 200021_169506), project PyroTRACH (ERC-2016-CoG) funded from H2020-EU.1.1. - Excellent Science - European Research Council (ERC), project ID 726165 and the European Union Horizon 2020 project FORCeS under grant agreement No 821205.

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
