# Peer review of "Molecular scale description of interfacial mass transfer in phase separated aqueous secondary organic aerosol"

_Atmospheric Chemistry and Physics, 2021_

## Author Response (AR1)

**Answers to the comments of Referee 2.**

This manuscript reports on the MD simulation results of water uptake on LLPS media containing an outer shell of organic material and an inner aqueous core. These simulations show a temperature dependence, indicating entropic factors are important, shows that water accumulation is energetically favored at the in the organic phase at the organic/aqueous interface at 300K. The uptake of water from the organic phase to the core is limited at 300K due to the entropic cost of the orientational ordering of water molecules. No such free energy barrier exists at 200K. Both diffusion effects and uptake effects can cause water to accumulate in the organic shell, with a concentration maximum at the core/shell interface, leaving the vapor/shell interface depleted in water and thus able to maintain a reduced surface tension. The authors take these observations and apply them generally to explore the influence of these effects on cloud droplet formation via Kohler curve simulations.

Overall, the paper is well-written, and the manuscript discusses topics of broad interest that are relevant to the readership of the journal. My expertise does not allow me to robustly comment on the specific details of the simulations, but assuming the results can be trusted, this manuscript presents a compelling molecular-level description for the effects of LLPS in CCN activation. My comments below relate primarily to the configuration of the system interrogated in the simulations, and how this is representative of real aerosol systems.

1) The authors define the interface-normal which is the path along which impinging gas phase water will follow to interact with the system. It is unclear to me if only trajectories following a surface normal were simulated, or if angled trajectories and glancing impacts were considered. Would these show different results to those presented? How much does the momentum of the molecule impact its adsorption into the system?

*Since it is only the interface normal coordinate of the pulled molecule that is biased by the steered MD simulation, the molecule is free to move in the XY directions parallel to the interface. This setup enables does not constrain the direction of the impact. To clarify this question, in Appendix A we added simulation snapshot series from three different trajectories that show different impact angles. The pulling velocity is chosen to be slow enough for the impact to resemble a scenario where a molecule moving with diffusive motion in the gas phase impacts a particle. This simulation setup thus closely mimics a real atmospheric impact event.*

2) There is a region of low density between the shell and the core, where both water and hCPA drop to 5% of their bulk densities. Does this imply there is a void in the system at this interface? Can the authors comment on and/or clarify this?

*We now explain in the text that density drop at the interfaces of two immiscible or partially miscible liquid phases - i.e. that the total density of the system changes through a minimum from the density of one bulk phase to the other - is commonly observed in molecular simulations and is attributed to the molecular level roughness of the interface (see for instance Jedlovszky et. al. J.Chem.Phys, 117 (5), 2002). It results in voids and a minimal distances between the molecules of the two phase which are larger than the average distances in the two bulk phases, but not a uniform layer of "vacuum" near the surface. The sign of these minimal distances between the phase on the left and the right can tell if the interfacial region is mixed or truly separated (Fernandes et. al. J. Phys. Chem. B 1999, 103, 8930-8939). Analysis clarifying the nature of the interface was added to Appendix A of the revised version.*

3) What factors determine the thickness of material required for it to appear "bulk-like"? The water shows bulk density within <0.5nm, but the hCPA does not show bulk density across the full 2 nm shell.

*The interface width determines what is the minimum width at which a phase appears bulk like. The density profile of the surface molecules follows a curve of Gaussian shape, and the overlap of the Gaussians determines whether a bulk is formed between the two sruface layers. The molecule size is trivially related to the width of these Gaussians.Layer by layer density profiles of the organic and the aqueous phase are added to Appendix A.*

4) What is the phase of water in the system at 200 K? In the atmosphere, this would presumably be ice, but there is no

mention here as to whether the water is liquid or solid. How would a solid ice core affect the observations / conclusions? Are these simulations actually representative of the phases of matter that would be encountered at 200 K?

*Despite of its more pronounced layered structure, the shape of the observed density profile at 200 K clearly resembles the liquid phase more than the density profile of ice. We calculated the diffusion coefficient of water from the 200 K simulation, and the obtained value ($10^{-13}$ $m^2$ $s^{-1}$), which suggests the core is in a semi-solid phase. The O:C of h-CPA is 0.4 and Tg(hCPA)/T $\sim$ 1, thus a real particle would be solid (or on the border of semisolid and solid) but phase separated, since LLPS always happens if O:C<0.56. A semi-solid phase separated particle is a valid assumption for the upper-troposphere (Atmos. Chem. Phys., 20, 8201–8225, 2020). This work aims to model the changes in mass accommodation coefficient of a the particle formed at a lower height (higher temperature) that rises towards lower temperatures. For particles formed at higher altitudes slow diffusion may prevent phase separation. Since $\alpha$-pinene is emitted near the surface, we assume that corresponding SOA would more likely form at lower heights. We acknowledge that 200 K is an extremely low temperature to examine, however it is common in molecular simulations to use exaggerated conditions in order to accentuate trends along variables such as temperature or pressure. Assuming a crystalline ice core is clearly a something to consider in a follow-up study, we hypothetise that it may affect the overall free energy difference between the condensed and the vapor phase, but - due to the existence of a quasi liquid layer between ice and the adsorbates (e.g.: Tasaki et al.,J. Phys. Chem. C 2008, 112, 7, 2618–2623) - the key behavior at the water/organic interface would presumably be very similar to what is observed in the present study. These considerations about the choice of the model system are now also included in the beginning of the Atmospheric Implications section.*

5) The presentation of data in figure 4(b) is somewhat confusing (I assume it's a bar graph, but this seems like a poor graph format when the bars for most of the data span the full height of the axis). Would suggest a data table in place of a graph here.

*Since, due to the changes in the manuscript vapor-to-water uptake coefficient have been removed, we removed the corresponding figure, and we list the values of the transport coefficients in text.*

**Answers to the comments of Referee 1.**

The present paper aims at obtaining microscopic understanding of water uptake into inhomogeneous aerosols. I understand the importance of this topics, and the questions posed in Introduction and summarized in Conclusions are very interesting. The authors adopted the steered MD to calculate free energy profiles. While the steered MD is proven to yield accurate and equivalent results of other methods based on equilibrium sampling in principle, the statistical accuracy should be carefully evaluated to confirm the convergence. My only and fundamental concern in this paper is related to the accuracy or the reliability of the results.

*The authors thank the referee for the careful and in detail examination of the methodological background and free energy profiles. Addressing the referee's comments have shed light on two technical errors and one typographical error in the text which account for most of the issues raised in this review:*

1. *We have a found that altogether 18 trajectories in the 300 K sample were impacted by an error we committed when we prepared the early versions of the Plumed input file, which caused double sampling of the aqueous phase. These trajectories were removed from the sample and free energy profiles were recalculated. Although this error did not affect the 200 K simulations, for consistency, 200 K profiles were also recalculated using the same number of samples (18 trajectories were removed randomly). The total number of simulations has been modified accordingly in the manuscript.*

2. *We also found an error in the calculation of the free energy profile using the Hummer-Szabo equation. The free energy profiles and all related quantities were recalculated with the corrected code.*

3. *The text in Appendix A erroneously stated that intrinsic density profiles are anchored to the water surface (a carry-over from an early draft). In the submitted manuscript, so called cross profiles are presented that is water density is anchored to the interface of the organic layer and vice versa.*

*Details of the free energy profiles have changed due to these corrections, values and locations of maxima and minima, as well as the vapor phase values have been slightly altered. So all analyses were repeated using the corrected profiles, and affected numbers and figures have been updated. Since the main features of the free energy profiles have been preserved, the conclusions have not changes due to these corrections. We acknowledge that a statistical analysis of the results is necessary, as steered molecular dynamics are not considered as benchmark methods in the field with a known statistical accuracy. We thus add a section to the Appendix (Section C) that contains statistical analysis and concludes that the given number of samples are sufficient to produce statistically significant results within the uncertainty that can be attributed to the energy of thermal motion at the temperature of interest.*

In the free energy profiles in Figure 2, I have some issues assessing the results. The following are the questions about the accuracy, and I wish the authors to address these questions.

    * Free energy profiles should take asymptotic values in both ends of vapor and water phases. (Otherwise, the system size is not sufficient.) The calculated profiles do not seem to show the asymptotic values. Is this behavior acceptable?

*The new free energy profiles do show asymptotic values in the bulk phases. Note that the extent of these asymptotic regions is defined by the endpoints of the steered MD trajectories.*

The difference in the asymptotic values should be consistent with the equilibrium vapor pressure of water. The author should confirm that the values are at least consistent with the vapor pressure in a semi-quantitative sense.

*We compare our simulated data to the free energy of transfer (evaporation) of water estimated from the experimental density of the condensed phase (liquid or supercooled water) and the partial vapor pressure (vapor phase density) of water. The calculations are described in Appendix C of the revised manuscript. Simulations tend to tendentiously underestimate the free energy of transfer of water, which can be partially attributed to limitations of the model potentials and partially to the limited availability of experimental data for our system of interest. In our case, the additional organic layer, the dissolved molecules in*

*the aqueous phase and the fact that the organic molecules are described by a basic transferable potentials further complicate direct and quantitative comparison. While quantitative agreement with experiments is clearly out of reach, temperature dependent trends are reproduced and simulated values are within 27 and and 10% from the free energies estimated from experimental data at both temperatures.*

The free energy profile in the water phase should be consistent with the density profile of water. This should be a consequence that the bulk water and the tagged water are identical.In relation to this issue, the structure of Delta G profile in the water phase (s = 0   2 nm) is hard to understand. I might suspect that this is a kind of artifact?

*The new free energy profiles do not have the feature mentioned in this comment. At this point we would like to emphasize that due to the non-neglibible miscibility of h-CPA with water, the point where the free energy profile terminates does not correspond to pure bulk water and contains a finite number of h-CPA molecules, thus free energy profiles - and especially free energy differences - differ from what is expected for pure water.*

I have also some questions about the usage of intrinsic surface analysis.

Since the system in this paper contains two interfaces, water-hpca and hpca-vapor, there are two ways to define the intrinsic coordinate. Which interface was used to define the z coordinate? Does the choice affect the result of the profile?

*We calculate the intrinsic density profiles using the liquid/liquid interface, such that each profile is anchored to the first molecular layer of the opposite phase. The collective variable in our simulations is independent of the choice of the surface to which intrinsic density profiles are anchored, thus free energy profiles are also independent of this choice. This has been clarified in the revised version of the manuscript. It is also possible to calculate profiles anchored to the liquid/vapor interface for the organic phase, however the density profiles of the aqueous phase anchored to an independent interface may be difficult to interpret due to the fact that the liquid/vapor interface fluctuates independently from the liquid/liquid surface.*

What is the merit of using the intrinsic coordinate in the present system? As far as I understand, the intrinsic coordinate results in an apparent layering structure of density profile, since it removes corrugation. However, Figure 2 does not show such apparent layering structure at 300 K. (Some structure is discernible at 200 K (Figure 2b), but I guess that liquid water at 200 K may not be realistic...)

*We use the intrinsic surface analysis not only to unravel the detailed structure of the density profiles **decoupled from the widening of the surface** caused by capillary wave corrugations which results in a smeared profile with no structure but also to separate molecules that constitute the interface for further surface selectuve analysis of structural properties. The density profiles calculated in this work are in good agreement with those seen in previous studies. At room temperature the aqueous phase is considerably less structured than the organic phase, (see Jorge et al. Journal of Physical Chemistry C, 114 (43). pp. 18656-18663), even for a truly insoluble counterphase (dichloromethane) only a first peak and a small second peak can be observed. Surface layering can be even less pronounced at the interface of partially miscible liquids than for truly immiscible phases. In the revised version, we present intrinsic mass density profiles of the aqueous phase compared to global non intrinsic ones in order to demonstrate that the intrinsic analysis does shed light on the fine structure of the surface and the subsurface regions.The reduced mobility at low temperature results in an enhancement of the structure, and hence more layers can be observed. We infer that the structure corresponds to supercooled liquid water that can be realistic in LLPS aerosol under atmospheric conditions, where the real aqeous phase is a multicomponent mixture of soluble organics and ions, a paragraph has been added to the Atmospheric Implications section of the manuscript to demonstrate the plausibility of such a phase. According to the latter, in future work we plan to address the effect of ionic strength in the aqueous phase on the resulting profiles and also to assess the effect of a frozen aqueous core. The additional merit of the intrinsic surface analysis is that it can unambigously determine the real surface molecules. Other traditional selection methods such as those relying on the 90-10% density definition of the interface can - due to their static nature - lead to the misidentification of bulk molecules as interfacial ones, leading to systematic errors of unknown magnitude any surface selective analysis (for instance: Pártay, L. B., Horvai, G.,  Jedlovszky, P. (2008). Molecular level structure of the liquid/liquid interface. Molecular dynamics simulation and*

*ITIM analysis of the water-CCl 4 system. Physical Chemistry Chemical Physics, 10(32), 4754-4764.). Interfacial molecules selected by the ITIM algorithm exactly represent the surface of the liquid phase at any moment in time, therefore any analysis of the structure therefore the intrinsic treatment removes the above mentioned systematic error from the results. In this work surface molecules are used to estimate orientational distributions and orientational entropies based on the first two molecular layers determined by the ITIM algorithm.*

Decomposing the free energy into energetic (enthalpic) and entropic contributions is known to be more challenging than the calculation of free energy itself. Usually, the decomposition requires more statistical sampling than the free energy calculation in one order of magnitude. Therefore, unless the authors confirm the statistical accuracy of the free energy, I could not evaluate the results of decomposition at present. The careful discussion on the accuracy of free energy calculation is required.

*We have demonstrated the statistical accuracy of the free energy calculations (Appendix C of the revised manuscript), and we have also showed and indicate in the text that both the free energy differences and the internal energy (enthalpy) values are in semi-quantitative agreement with experimental free energies of transfer and latent heats of vaporization for pure water at the simulated temperatures. Nevertheless, since we could not provide experimental comparison for the total entropies, we agree to remove the total entropy profile. The information that the free energy profiles have distinct features and enthalpy profiles are smooth functions is sufficient to conclude that features of the free energy profile have entropic origins. In the revised version only distinct contributions to the total entropy are kept and analysed in relation to their effect on the free energy profile. Figure 3 has been changed accordingly.*